



**Spatiotemporally resolved emissions and concentrations of Styrene, Benzene,**
**Toluene, Ethylbenzene, and Xylenes (SBTEX) in the U.S. Gulf region**
Chi-Tsan Wang[1], Bok H. Baek[1*], William Vizuete[2], Jia Xing[1], Jaime Green[2], Marc Serre[2], Richard
Strott[2], Lawrence S. Engel[3], Jared Bowden[4], and Jung-Hun Woo[5]
[1]Center for Spatial Information Science and Systems (CSISS), George Mason University, Fairfax,
VA, U.S.
[2]Department of Environmental Sciences & Engineering, University of North Carolina, Chapel
Hill, NC, U.S.
[3]Department of Epidemiology, University of North Carolina, Chapel Hill, NC, U.S.
[4]Departement of Applied Ecology, North Carolina State University, Raleigh, NC, U.S.
[5]Civil and Environmental Engineering, College of Engineering, Konkuk University, Seoul,
Republic of Korea
[*]Corresponding authors: Bok H. Baek (Email: bhbaek@gmail.com; Telephone: +1 919-308-6519)
**Abstract**
Styrene, Benzene, Toluene, Ethylbenzene, and Xylenes (SBTEX) are established neurotoxicants.
These SBTEX are hazardous air pollutants (HAPs) and released from the petrochemical industry,
combustion process, transport emission, and solvent usage sources. Although several SBTEX toxic
assessment studies have been conducted, they have mainly relied on ambient measurements to
estimate exposure and limiting their scope to specific locations and observational periods. To
overcome these spatiotemporal limitations, an air quality modeling system over the U.S. Gulf
region was created predicting the the spatially and temporally enhanced SBTEX modeling
concentrations from May to September 2012. Due to the incompleteness of SBTEX in the official
US EPA National Emission Inventory (NEI), Hazardous Air Pollutions Imputation (HAPI)
program was used to identify and estimate the missing HAPs emissions. The improved emission
data was processed to generate the chemically-speciated hourly gridded emission inputs for the
Comprehensive Air Quality Model with Extensions (CAMx) chemical transport model to simulate
the SBTEX concentrations over the Gulf modeling region. SBTEX pollutants were modeled using



a "Reactive Tracer" feature in CAMx that accounts for their chemical and physical processes in
the atmosphere. The data shows that the major SBTEX emissions in this region are contributed by
mobile emission (45%), wildfire (30%), and industry (26%). Most SBTEX emissions are emitted
during daytime hours (local time 14:00 -17:00), and the emission rate in the model domain is about
20 - 40 t $hr^{-1}$, which is about 4 times higher than that in the night-time (local time 24:00 – 4:00,
about 4 – 10 t $hr^{-1}$). High concentrations of SBTEX (above 1 ppb) occurred near the cities close to
the I-10 interstate highway (Houston, Beaumont, Lake Charles, Lafayette, Baton Rouge, New
Orleans, and Mobile) and other metropolitan cities (Shreveport and Dallas). High Styrene
concentrations were co-located with industrial sources, which contribute the most to the Styrene
emissions. The HAPI program successfully estimated missing emissions of Styrene from the
chemical industry. The change increased total Styrene emissions was increased by 22% resulting
in maximum ambient concentrations increasing from 0.035 ppb to 1.75 ppb across the model
domain. The predicted SBTEX concentrations with imputed emissions present good agreement
with observational data, with a correlation coefficient (R) of 0.75 (0.46 to 0.77 for individual
SBTEX species) and normalized mean bias (NMB) of -5.6% (-24.9% to 32.1% for individual
SBTEX species), suggesting their value for supporting any SBTEX-related human health studies
in the Gulf region.
**Keywords:** BTEX, Styrene, SMOKE, Reactive Tracer, Toxicants, HAP, CAMx, Exposure



## 1. Introduction


Styrene, Benzene, Toluene, Ethylbenzene, and Xylene (SBTEX) are listed as Hazardous Air
Pollutants (HAPs) by the U.S. Environmental Protection Agency (EPA) (Declet-Barreto et al.,
2020) and can be detected in unhealthy amounts in the ambient environment. The SBTEX is
primarily from industrial emission sources and can be found in the petrochemical, construction,
and manufacturing industries (Polvara et al., 2021; Declet-Barreto et al., 2020) with 98% of
benzene emissions attributed to coal and petroleum sources (ATSDR, 2007a, b, 2010a, b, 2017).
Exposure studies of total SBTEX at industrial sources in the Middle East, Europe, and West Asia,
have shown that workers experience a cumulative yearly environmental exposure of 25 - 176 ppb
(Al-Harbi et al., 2020; Rajabi et al., 2020; Christensen et al., 2018; Rahimpoor et al., 2022; Niaz
et al., 2015; Moshiran et al., 2021). The inhalation reference concentration for Benzene shows
low-dose linearity utilizing maximum likelihood estimate E-5 risk level of Benzene (1 in 100,000)
range is 0.4 -1.4 ppb of air concentration for leukemia (USEPA, 2000).
Given the importance of SBTEX from industrial sources, the heavily industrialized Gulf region of
the U.S. could be a significant source of exposure for the population living there. According to the
Agency for Toxic Substances and Disease Registry (ATSDR) report, the petrochemical industry
in the Gulf region states contributes approximately 52% (~5.3 million tons yr$^{-1}$) of Benzene
production capacity in the U.S. (ATSDR, 2007a) and ~75% (~6.2 million tons yr$^{-1}$) of Xylenes
production capacity (ATSDR, 2007b). Texas and Louisiana have significant production of Styrene
and Ethylbenzene, with an annual production of 5.5 and 7.2 million tons yr$^{-1}$, respectively (SRI,
2008; ATSDR, 2010a). A recent study of SBTEX exposures in the U.S. Gulf region, conducted
within the Gulf Long-term Follow-up Study (GULF Study) cohort (NIEHS, 2021), observed
associations of blood concentrations and annual average air concentrations of these chemicals with
neurological symptoms (Werder et al., 2019; Werder et al., 2018). The average blood BTEX
concentration among the 146 tobacco smoke-unexposed participants with blood measurements in
this study was 255 ng L$^{-1}$ es (Doherty et al., 2017; Werder et al., 2018) . This value is similar to
that for a representative nationwide sample assessed as part of the US National Health and
Nutrition Examination Survey (NHANES) in 2005-2008 (NCHS, 2021), which measured an
average of 247 ng L$^{-1}$. In the GuLF Study population study, however, the 95$^{th}$ percentile of BTEX
concentrations was 991 ng L$^{-1}$, which is 23% higher than the 95$^{th}$ percentile for the NHANES
nationwide sample of 803 ng L$^{-1}$. The mean blood concentration of Styrene for the GuLF Study





sample was 52 ng L$^{-1}$ (95$^{th}$ percentile: 882 ng L$^{-1}$), or twice the NHANES nationwide mean of 25
ng L$^{-1}$ (95$^{th}$ percentile: 55 ng L$^{-1}$) (NCHS, 2021). Due to the short biological half-lives of SBTEX
species, the study concluded that this high average SBTEX concentration in blood in the Gulf
Region resulted from recent, presumably local emission sources.
Most ambient exposure studies of SBTEX have relied directly on local measurements from the
field, or at existing ambient monitors. These measurements can then be used in statistical models
to spatially predict exposures to SBTEX (Pankow et al., 2003; O'Leary and Lemke, 2014; Miller
et al., 2018; Hsieh et al., 2020b). For example, Hsieh et al. (Hsieh et al., 2020a) developed the
Multivariate Linear Regression (MLR) models to estimate SBTEX concentrations using
correlations with other criteria air pollutants, including nitrogen oxides (NOx), carbon monoxide
(CO), sulfur dioxide (SO$_2$), particulate matter (PM), and meteorological conditions (temperature,
wind speed). The MLR model predicted a strong correlation with NOx and CO. The limitations of
the statistical model are that they require measurement data, and they assume that the
measurements originate from a single source in a relatively small region. The use of a dispersion
model is another way to estimate ambient SBTEX concentrations when local measurements are
lacking. Chen et al., 2016 (Chen et al., 2016) applied a dispersion model to predict SBTEX and
other toxicant concentrations in two industrial complexes in Kaohsiung City, Taiwan. The
dispersion model performed better for stationary point sources than a statistical based model and
predicted up to ~78% of the ambient observation. These dispersion models, however, account for
exposures at a small spatial temporal scale and cannot support regional scale application.
Furthermore, these models assumed that the exposure rate to SBTEX is linear, without considering
any chemical destruction and wet/dry deposition losses in the atmosphere.
An accurate SBTEX assessment in the Gulf region must address the known uncertainties
associated with current statistical, biometric, and dispersion model approaches. Improved accuracy
in exposure estimation is dependent on the inclusion of all industrial emission sources, must
capture the temporal and spatial variability known to occur in industrial emission rates, and should
include the chemical and physical decay processes of the atmosphere. These issues can be
addressed using a regional-scale chemical transport model (CTM), like the Comprehensive Air
Quality Model with Extension (CAMx) (RAMBOLL, 2021) coupled with an emission inventory
with a comprehensive accounting of all SBTEX sources. Because the current CAMx model





simulation process cannot support SBTEX simulation with reduced chemical mechanism, the post
process called reactive tracer function is used to overcome the limit of the reduced mechanism.
Currently, SBTEX emission data can be found in EPA's National Emission Inventory (NEI),
which includes data from the Toxics Release Inventory (TRI) program database (USEPA, 2021a).
Unlike for benzene sources, the TRI data for the other four species (STEX) is based on voluntary
reports, and as a result, the 2011 NEI has emission rate data for these air toxics only for a limited
set of emission sources (USEPA, 2021d).
The following work describes the development of a new STEX emission inventory for the Gulf
Coast region that includes the emission sources absent from the 2011 NEI. Missing emission rate
data of STEX was provided by analyzing NEI emissions of similar industrial sources that did
provide emission rates and applying their rates to the missing source. Diurnal profiles for STEX
were based on the hourly profiles of other pollutants with the same type of industrial source. This
study then applied the Sparse Matrix Operator Kernel Emissions (SMOKE) model system (Baek
and Seppanen, 2021) to generate a CAMx-ready emission inventory. Since STEX are not included
as explicit species in the chemical mechanisms used by CAMx, a reactive tracer was included to
account for chemical losses. This new emission inventory was then utilized in CAMx to predict
STEX concentrations over the Gulf region for 5 months in 2012.



## 2. Materials and Methods

Benzene emission reporting is mandatory in the NEI and thus was assumed to be comprehensive. Only the STEX portion of the inventory, with voluntary reporting, was the focus of the investigation for missing sources. The emission inventory used as a base was the 2011 version 6 NEI. Missing emission sources were then added to that inventory relying on information from TRI (USEPA, 2021a) and Emission Inventory System (EIS) (USEPA, 2022b). The SMOKE modeling system was then used to generate the STEX hourly gridded emissions over the Gulf modeling region for 2012.

### 2.1 Emission Data Preparation

### 2.1.1 The HAPs data in NEI Emission Inventory

The emissions data collected from certain facilities by all state agencies responsible for regulating air pollution are submitted to the USEPA by EIS (USEPA, 2022b). They use them to develop the (NEI). The NEI is a national database of comprehensive estimates of annual air emissions of criteria air pollutants (CAPs) (e.g., C.O., $NO_X$, $SO_2$, $NH_3$, VOC, and $PM_{2.5}$), and HAPs (e.g., benzene, acetaldehyde, formaldehyde, xylenes, Styrene, and more) from all types of emissions sources (e.g., point, nonpoint, and mobile). While the CAPs reporting by the agencies is mandatory, reporting HAPs is voluntary. Thus, only limited HAPs have been reported to the USEPA, and their spatial coverage can vary significantly by source type (e.g., industrial, vehicles) and region (e.g., county and state) (Strum et al., 2017).

The VOC emission species in NEI have three types, "model surrogate", "model explicit", and "HAPs explicit" species. The "model surrogate species", such as XYL (Xylene and other poly-alkyl aromatics), TOL (Toluene and other mono-alkyl aromatics), and PAR (paraffin carbon bond), are used to predict ozone in the CTM but not for individual HAPs emission and simulation. Only five HAP emissions in NEI are "model explicit" specie: naphthalene (NAPH), Benzene (BENZ), acetaldehyde (ALD2), formaldehyde (FORM), and methanol (MEOH), known as "NBAFM" to represent their individual emission (Strum et al., 2017), and are directly processed in CTM model, too. The "HAPs explicit" species emission in NEI includes hundreds of toxicants (such as Styrene, Xylenes, Mercury, and Acrolein). Those "HAPs explicit" species cannot be



directly used in the current CTM model because their explicit chemical mechanisms are not
developed in the current CTM chemical mechanism.
The "model explicit" species, Benzene (B), and other "HAPs explicit" species, including Styrene,
Toluene, Ethylbenzene, and Xylenes (STEX) are targeted for this SBTEX human exposure study.
The SMOKE model system (Baek and Seppanen, 2021) assigned the annual or monthly SBTEX
emission inventory in NEI to hourly emission patterns by the temporal profiles based on emission
processes and locations by Source Category Code (SCC) and Federal Information Processing
Standards (FIPS) county codes. These processes are coupled with the CAPs when generating the
CTM-ready emission data.

### 2.1.2 Imputation of NEI with STEX

This study utilized the 2011 NEI summary reports from the SMOKE modeling System (Baek and
Seppanen, 2021) to identify those missing STEX emission sources. The SMOKE reports provided
the annual or monthly total of VOC and individual HAPs emissions sorted by SCC and FIPS
county codes. This study developed an R-project (The R Foundation, 2021) program called
"*Hazardous Air Pollutants Imputation*" (HAPI) that can first read the reports from SMOKE and
identify the list of inventory sources reported without STEX toxics. Then it generates the
imputation data for those missing STEX inventory sources based on the proxy of STEX and VOC
for those emission sources that share the same SCC near the region (county or state).
Theoretically, the SCC is the reference code defining the emission process type. The same SCC
means they share similar emission factors with the same emission process (USEPA, 2016). The
profiles of HAPs for the VOC can be shared with those same SCC emission sources within the
surrounding regions (counties or states)(Strum et al., 2017). When there are the same SCC
emission sources with zero HAPs in other counties, this study performed the imputation of those
missing HAPs emissions based on the HAPs profiles from the matched emission source. For
example, the HAPs profile of Styrene and Toluene to the VOC emission is defined as the ratio of
Styrene and toluene emissions over the VOC emission ($P_{toluene,s}$) in counties where there are
Styrene, Toluene, and VOC emissions for that SCC ($s$). Then, this study will assume that those
HAPs are missing when the summation of HAPs emissions are zero ($\sum_i E_{i,s,f} = 0$: $i$ is pollutants,



$s$ is SCC code, $f$ is FIPS county code) but VOC emission is available. Then this will apply the
HAPs profile for the same SCC to the existing VOC and estimate missing Styrene and toluene
emissions. Therefore, this process can impute the missing HAP emissions based on the SCC-
matched HAPs fractions from the surrounding counties or the same state.
The HAPI was developed based on this imputation concept. This study first separated the county
and SCC level inventory data into two groups in the HAPI program: "with HAPs" and "without
HAPs." For the "with HAPs" group, summations of HAPs emissions in counties and SCCs are not
zero. In contrast, for the "without HAPs" group, summations of HAPs emissions in counties and
SCC are zero.
In the "with HAPs" group ($\sum_i E_{i,s,f} > 0$) in Eq. (1), $i$ is the individual HAP, such as Styrene,
Benzene, Toluene, Ethylbenzene, xylenes, acrolein, and 1,3-butadiene; $s$ is the SCC, and $f$ is the
county FIPS code for county. $E_{i,s,f}$ is the annual emission of pollutant $i$ for SCC in the county. $E_{voc}$
is the CAP VOC emission for the SCC in the county. The HAPs profile ($P_{i,s}$) is a fraction of HAP-
specific emission ($E_{i,s,f}$) over the summation of matched SCC and county-specific VOC emission
($E_{voc,s,f}$) from the "with HAPs" group.
This study assumed that if there is an SCC-matched "with HAPs" group HAPs profile in the
inventory, they are not considered as missing HAPs emission sources. Only the emission sources
with the sum of all HAPs are zero considered as "without HAPs" group ($\sum_i E_{i,s,f} = 0$). In Eq.(2),
$P_{i,s}$ is used to estimate those missing HAPs for the "without HAPs" inventory source group. The
$E_{voc,s,f}$ is the CAP VOC emission in the "without HAPs" group.
When $\sum_i E_{i,s,f} > 0$, calculate individual HAPs to total VOC ratio ($P_{i,s}$):

$$P_{i,s} = \frac{\sum_f E_{i,s,f}}{\sum_f E_{voc,s,f}}$$

203                                                                                          Eq. (1)

When $\sum_i E_{i,s,f} = 0$, the HAPs emission are missing, this study applied $P_{i,s}$ and VOC emission to
estimate the missing HAPs emission:
$$Em_{i,s,f} = P_{i,s} \times E_{voc,s,f}$$                                   Eq. (2)
The HAPI program then outputs the total HAPs emissions ($Em_{i,s,f}$) for the SMOKE modeling
system to integrate with the CAP VOC inventory described in Section 2.1.2. Finally, the HAPI



program performs the quality assurance step again to confirm that there are no missing HAPs after
imputation and that the summation of HAPs emissions is not greater than the CAP VOC emission.

## 2.2 Model Configuration

After developing the CTM-ready emissions, the Comprehensive Air Quality Model with Extension
(CAMx, version 7.0) (RAMBOLL, 2021) with the "Reactive Tracer" (RTRAC) post-process feature
was used to simulate the ambient SBTEX concentration over the Gulf region. The year 2012
Weather Research and Forecasting (WRF) simulated meteorology data were developed by USEPA
Support Center for Regulatory Atmospheric Modeling (SCRAM) (USEPA, 2022a). They were
converted to SMOKE- and CAMx-ready gridded hourly meteorology through the Meteorology
Chemistry Interface Processor (MCIP). The meteorology-induced emissions sectors, such as
onroad (Choi et al., 2014; Lindhjem et al., 2004) and biogenic, are estimated with the MCIP
gridded hourly meteorology. The USEPA's daily total wildfire emissions (ptfire) estimated by
SMARTFIRE2 (USEPA, 2015) were imported for the year 2012 emissions modeling (USEPA,
2021b). The base air quality model descriptions and evaluations are in the supplementary
document. The overall research method scheme flowchart is shown in Fig. S1.
The "Reactive Tracer" is a post-analysis feature in the CAMx modeling system to simulate SBTEX
concentrations. Along with the physical decay processes like wet and dry deposition, there is the
second-order chemical reduction rate $r$ that is calculated using the oxidants (Ozone, OH, $NO_3$)
concentrations $[Ox]$, the SBTEX concentrations $[Tr]$, and the rate constants of reactions $k_{Tr+Ox}$
(Eq.3). In Eq.4, $k$ is the rate constant calculated by $A$, $B$, temperature ($T$), and activation energy
($E_a$). The Master Chemical Mechanism for aromatic schemes (Bloss et al., 2005) is considered for
the parameters of each specific reaction in the RTRAC process.
This study considered the initial reactions of SBTEX in the MCM mechanism version 3.3.1 (Jenkin
et al., 2015). For other parameters, the National Institute of Standards and Technology (NIST)
Chemistry Webbook (P.J. Linstrom and W.G. Mallard, 2018), and CAMx user guide (Ramboll,
2020) are considered for determining the Henry's Law constant, dependence temperature, and
molecular weight. All parameters used in our RTRAC modeling are presented in Tables S3 and
S4.



$$r = k_{Tr+Ox} [Tr][Ox] \qquad \text{Eq. (3)}$$
$$k = A(\frac{T}{300})^B \exp(\frac{-E_a}{T}) \qquad \text{Eq. (4)}$$

### 239 2.3 Ambient SBTEX Measurements

The CAMx modeling evaluation was completed with the USEPA Air Quality Station (AQS) ozone
observational data and the Texas Commission on Environmental Quality (TCEQ) State
Implementation Plan (SIP) ozone modeling output data (TCEQ, 2015). The measured ambient
SBTEX concentrations are from the USEPA Ambient Monitoring Technology Information Center
(AMTIC), which is an observational network that routinely detects more than 100 air toxics in the
U.S. (USEPA, 2021c). It includes the federal and state monitoring stations. The long-term
individual SBTEX concentrations from the AMTIC were utilized to evaluate the RTRAC
modeling results from CAMx.
A total of 46 monitoring sites measure SBTEX concentrations within our 4 km × 4 km model
domain, and most of them are located within Texas (42 sites), except for four sites in Louisiana.
The air sampling duration can be 1-hour, 3-hour, or 24-hour. There are six monitoring sites with
1-hour measurement data in Texas, three sites with 3-hour data in Louisiana, and the rest with 24-
hour data. The AMTIC sites are indicated in Fig. 1 with red stars. This study applied twice the
interquartile range (2*IQR) above Q3 to remove the observational outliers that can be captured by
the monitoring sites. Those outliers are usually caused by unpredictable high/low concentration
events (Couzo et al., 2012).
The CAMx RTRAC modeling results are spatially and temporally resolved gridded hourly
concentrations, while the AMTIC observational data are from specific locations with time gaps.
Daily average and diurnal pattern analyses evaluate the predicted SBTEX concentrations. For each
AMTIC site, this study used the average concentration of the center grid cell and eight other
"surrounding" grid cells (i.e., the average of 3×3 grid cells) to compare with the observational data
(USEPA, 2006).



## 3. Results

### 3.1 SBTEX Emissions

The 2012 annual total SBTEX emissions in the model domain are shown in Table 1. The emission sectors include: agriculture fire (afgire), commercial marine vehicle (cmv), non-point source (nonpt), non-road vehicle (nonroad), on-road vehicle (onroad), fire emission (ptfire), rail road (rail), residential wood combustion (rwc), non-point oil gas industry (np_oilgas), electricity power plants unit (ptegu), point source emission other than electricity generation unit (ptnonipm), and point source of oil and gas industry (pt_oilgas). The largest contributor of SBTEX emissions in the 12km×12km model domain is indicated to be from the "onroad" sector, with 89,204 t yr$^{-1}$, representing about 36% of the total SBTEX emissions. The "onroad" sector contributes the most of total Xylenes (46%), Toluene (48%), and Ethylbenzene (60%) emissions, while much less to Benzene (13%) and Styrene (6.8%). The second largest contributor of SBTEX emissions is the "wildfire" sector (61,316 t yr$^{-1}$), contributing about 25% of total SBTEX. The wildfire contributes the most of total Benzene (57%), 12% of total Toluene and 7% of total Xylene, but no Ethylbenzene and Styrene due to missing explicit profiles in the 2012 wildfire emission inventory. The "nonroad" sector ranked third (35,375 t yr$^{-1}$), contributing about 14% of total SBTEX over our modeling region. The nonroad contributes largely to Xylenes (15%), Toluene (21%), and Ethylbenzene (21%). Compared to other sectors, emissions from non-electricity generation unit industrial point sources (ptnonipm) contain a larger portion of Styrene, 2,911 t yr$^{-1}$, which is 69% of total Styrene emission. Our study successfully identified missing Styrene emissions from the chemical industry process (see table S7), leading to a 34% increase in total Styrene emissions.

The individual and total SBTEX annual emission spatial plots in 12km×12km model domain are presented in Fig. 2. The grid cell with the highest SBTEX emissions is found in Houston city near the ship channel (1059 t yr$^{-1}$), which is about 35 times higher than average emission (28 t yr$^{-1}$) across the domain, followed by one in San Antonio in Texas (1022 t yr$^{-1}$) and one near Sabine Lake in Louisiana (1022 t yr$^{-1}$). In Fig.2 (b), the missing sources of SBTEX emissions in the NEI are mostly located in Texas and Louisiana, particularly for the grid cells in Lake Charles (increased by 373 t yr$^{-1}$, +282%), Baton Rouge (167 t yr$^{-1}$, +31% ) in Louisiana; Belton (61 t yr$^{-1}$, +21%), Fort Worth (50 t yr$^{-1}$, +85%), Dallas (44 t yr$^{-1}$, +52%) in Texas, and some rural area in Texas.



These missing sources of SBTEX are mostly from the np_oilgas and ptnonipm emission sectors
(detailed in Supplementary document 3.1 and 3.2). Although the total SBTEX emission increased
by only 2% based on the domain average (Table 1), the localized impacts for certain areas can be
up to 60% of the total SBTEX emissions.
The SBTEX emissions exhibit strong diurnal variations across a day, as presented in Fig. 3a. The
daytime hourly emission (up to 77 t hr$^{-1}$) is about 4.3 times higher than the night-time emission
rate, mainly due to the larger emissions from on-road and off-road mobile sources (half of total
emissions) during the daytime. The diurnal variations in the chemical composition of total SBTEX
also suggested the increased percentage of Toluene and Xylenes (indicating the transport sources)
during the morning (L.T. 6:00 – 10:00) and evening (L.T. 19:00) rush hour. The inclusion of the
missing sources will slightly mitigate the emission variation across a day, as most of the missing
sources come from industrial manufacturing and oil processes (detailed in SI) whose diurnal
profiles are much flatter (about only 20% increase during the daytime) compared to the total
emission (see Fig. 3b), with much smaller differences between day (0.86 t hr$^{-1}$) and night (0.69 t
hr$^{-1}$). The chemical composition of missing emission sources were relatively constant throughout
the day with about 50% comprised of Xylenes, 30% Toluene, and Styrene was 10-15%. The
relative amount of missing Styrene was higher than that found in total emissions.
**3.2 SBTEX Concentrations**
CAMx simulations predicting SBTEX concentrations were completed using two sets of emissions:
the National Emission Inventory (Base), and the emission scenario adjusted in this study (Adj).
The differences between the two scenarios can be regarded as the impacts of the missing emission
sources in the original NEI, suggesting the importance of the completeness of emissions.
**3.2.1 Spatial Distribution**
Fig. 4a presents the spatial distribution of SBTEX concentration during the model period (May
1st to Sep 30th) in the Adj scenario. The highest SBTEX concentration (3.07 ppb) occurs near
Lake Charles, followed by Baton Rouge (2.06 ppb), Houston ship channel (2.04 ppb), Shreveport
(1.69 ppb), Beaumont (1.59 ppb). The individual SBTEX shows similar spatial distribution
patterns as they share similar emission sources except for Styrene. Because the main emission





source of Styrene is ptnonipm, while other species are mostly from vehicle emissions and wildfire.
Houston exhibits the highest concentration of Benzene (max: 1.06 ppb), Toluene (max: 1.01 ppb),
and Ethylbenzene (max: 0.16 ppb), corresponding to its large amount of SBTEX emissions, while
Xylenes (0.78 ppb) is from Shreveport. The highest concentration of Styrene (1.97 ppb) occurs
near Lake Charles where has abundant non-EGU point sources that have been missing in original
NEI emissions.
We further investigated the influence of missing emission sources in the original NEI on the
SBTEX concentrations by taking the differences between Adj and Base scenarios. The majority of
missing emissions are associated with the np_oilgas and ptnonipm sectors, with increased
contributions geographically concentrated in Texas and Louisiana (Fig. 4b). In particular, the
largest impact on SBTEX concentration is shown near Lake Charles by up to 1.82 ppb (+68%),
which is mostly related to the increase of Styrene concentration (by 1.75 ppb, +5315%). This
increase is due to the NEI missing one large point source (364.12 t yr$^{-1}$) in the ptnonipm sector
near Lake Charles. The inclusion of missing emission sources also led to the increase of Styrene
concentrations in other cities, such as Baton Rouge (0.07 ppb, +389%), LA, and Houston, TX
(0.03 ppb, +62%). Baton Rouge, LA also suffers the largest increase of Toluene concentrations by
0.44 ppb (+92%) due to the inclusion of missing emissions, followed by Beaumont (0.07 ppb,
+50%), and Carthage (0.048 ppb, +66%) in TX. Fort Worth, TX exhibits the most increase of
Xylenes concentrations by 0.07 ppb (+95%), followed by Center (0.06 ppb, +273%), Teague (0.06
ppb, +340%), and Beaumont (0.036 ppb, +70%) in TX. The largest increase of Ethylbenzene
concentration occurred at Longview (0.01 ppb, +85%), followed by Beaumont (0.009 ppb, +40%)
and Houston (0.006 ppb, +9%) in TX.
**3.2.2 The Diurnal Variation**
In general, the diurnal variations of SBTEX concentrations are mostly driven by meteorological
factors (e.g., ventilation, radiation), exhibiting lower concentrations in the daytime than night due
to stronger ventilation, and chemical loss, although the emission is higher during daytime than
night as we presented previously (Fig. 3). Diurnal meteorological and emission patterns suggest
more sensitivity to the concentration of emissions at nighttime than daytime, implying that
emission controls to reduce the concentrations at night would be most effective. The variation of





emission sources might also modulate the diurnal pattern in concentrations. To demonstrate that,
here we selected two industrial locations and one city location with high SBTEX concentrations
to compare the diurnal variation of concentrations.
The first one is Channelview city (Latitude: 29.8, Longitude: -95.12), located at the Houston ship
channel industrial area on the eastern side of downtown Houston. Driven by both emission
temporal profiles and meteorological conditions, the peak SBTEX concentration (about 12 ppb) in
Channelview city occurs at LT 23:00 to 1:00, contributed mostly by Benzene (56%) which
indicates the industrial sources, with a small amount of Toluene (19%), Xylene (13%), Styrene
(4.8%), Ethylbenzene (7%) (Fig. 5a). In contrast, the Bayland Park (Latitude: 29.69, Longitude: -
95.49) located nearby at the western side of Houston, presents the same level of peak SBTEX
concentration (about 12 ppb) (Fig.6a) as Channelview city. Different from Channelview city,
however, the peak concentration of Bayland Park occurs at traffic rush hour (LT 7:00 to 8:00),
contributed mostly by Toluene (53%) and Xylene (23%) (indicating the mobile vehicle sources)
rather than Benzene (18%). Meanwhile, the adjusted industry emission sources, which is present
in table S5, significantly contribute to the peak concentration (0.4 ppb) in Channelview city (Fig.
5b), but much less in Bayland Park (Fig.6b), which is far from the industry area.
A similar pattern is also shown in Baton Rouge, Louisiana (Latitude: 30.46, Longitude: -91.17),
located near downtown Baton Rouge (affected by onroad sources), and also close to the industry
area (~ 1 mile from the north). Like Houston industry area, the daytime SBTEX concentration is
much lower (<3 ppb) than night-time, and the peak SBTEX concentration (about 9.4 ppb) occurs
at LT 22:00 (Fig. 7). Because Baton Rouge is impacted by both traffic and industrial sources,
emissions differ from Houston in that both Benzene (35% – 40%) and Toluene (35% - 40%)
become the major portion of SBTEX (Fig.7a). The missing emission sources (Fig.7b) will further
enhance the peak concentration by 2 ppb at LT 5:00 - 8:00, with the largest chemical composition
of Toluene (about 70 - 85%), followed by Styrene (about 7 - 20%) associated with the industrial
sources.
**3.3 Comparison with Observations**



The simulated concentrations were compared with the observations to evaluate the accuracy of
SBTEX emission and concentration estimated in this study. The normalized mean bias (NMB, %)
and correlation coefficient (R) of both Base and Adj cases were compared in Table 2. Overall, the
CAMx model can capture the pollution level and spatiotemporal variation of all SBTEX species.
More specifically, the model reproduced the daily variation of SBTEX concentrations, with R of
0.65 (0.54-0.65 for individual SBTEX) for all daily observational records (N=2,717), as well as
their spatial distribution across observational sites (N=46, averages of the whole simulation
period), with R of 0.75 (0.46 to 0.77 for individual SBTEX species), and NMB of -5.6% (-24.9%
to 32.1% for individual SBTEX species).
The model also reproduced the diurnal variation of SBTEX concentrations as presented in Fig. S8
(three site data in Houston city). Additionally, the inclusion of emissions can slightly improve the
overall model performance with decreased NMBs for Toluene (+3.5%), Xylenes (+5.7%),
Ethylbenzene (+3.8%), and total SBTEX (+3.2%). The NMB for Styrene is increased from 17.4%
to 32.1%, while R is increased by 0.01, suggesting better correlations with the new-estimated
emission data, while uncertainties associated with emission factors or other parameters lead to the
overestimation of SBTEX. Fig. 8 shows the spatial distribution of average concentration simulated
in the Adj case, overlapping the average observational data for total SBTEX (8a) and individual
species (8b to 8f). The observational data (diamond shapes) shows a high concentration in
industrial or city sites, and a lower concentration at rural sites. The model results showed a
continual concentration gradient pattern from cities to a rural area with 4 km × 4 km resolution
and the results are close to the observational data in Houston, Dallas, Beaumont, and Baton Rouge.
We further classified the observation sites into four groups, including "Airport", "Industry",
"Rural", and "Urban" based on their geographical locations (Table S8). For total SBTEX (Fig. 9a),
the correlation coefficient (R) is 0.75 (R-square is 0.56) across all locations, and the black solid
line is the regression line for all sites (N=46). The red dots indicated that the industrial sites have
a higher concentration in both model and observational results, and the cities (blue diamonds)
showed that their concentrations are slightly overestimated and lower than industrial sites. The
Airport (black squares) and Rural (green triangles) have lower SBTEX concentrations than City
and Industry, and Rural is the lowest group. Fig. 9b to 9f are similar plots for explicit Benzene,
Toluene, Xylenes, Ethylbenzene, and Styrene. The R ranges from 0.46 to 0.77. The Benzene (R is



0.68), Toluene (R is 0.46), and Styrene (R is 0.64) are overestimated, but Xylenes (R is 0.77),
Ethylbenzene (R is 0.77) are close to observational data. Although Toluene has the lowest R (0.46),
which is caused by two industry sites that largely underestimate in Houston (Site ID: 482011015)
and Nederland (Site ID: 482450014), in case we remove those two industrial sites data, the R for
Toluene in Fig. 9c will become 0.7 (Fig. S9). This phenomenon is probably caused by the missing
Toluene industrial sources near those two sites. The inclusion of missing emission sources
definitively improved the model performance (Table 2), especially in Rural (+5.4%) and Airport
groups (+6.8%) which suffegred the most due to the missing industrial sources. The NMBs for
Xylenes are also reduced across all emission groups (Industry: +3%, Urban: +12%, Airport: +20%,
and Rural: +13%).
**4. Conclusion**
To address the urgent need for health assessment of SBTEX exposures in the Gulf region, this
study developed high spatiotemporally resolved emissions and concentrations of individual
SBTEX. We developed and implemented the HAPI program to identify and gap-fill the missing
SBTEX inventory for the SMOKE emissions modeling system. Then we successfully
implemented the state-of-the-science chemical transport modeling system, CAMx, to generate the
high temporal and spatial resolution predictions of explicit SBTEX concentrations based on the
improved SBTEX emission inventory and a "Reactive Tracer" (RTRAC) feature. The modeled
average SBTEX concentrations exhibit good agreement with observational data (R is 0.75 and
NMB is improved in Adj case to -5.6% for total SBTEX), suggesting that the emissions and
concentrations estimates developed in this study can be used to support well the SBTEX-related
human health studies in the Gulf region.
We also found that the "onroad" sector contributes the most to total Xylenes (46%), Toluene (48%),
and Ethylbenzene (60%) emissions, while the Styrene emissions are mostly contributed by non-
EGU point sources (ptnonipm, 69%) but were substantially missed in the original NEI data,
resulting in 34% underestimation of total Styrene emissions. The highest SBTEX concentration
(3.07 ppb) occurs near Lake Charles, followed by Baton Rouge (2.06 ppb), Houston ship channel
(2.04 ppb), Shreveport (1.69 ppb), Beaumont (1.59 ppb), corresponding to a large amount of
SBTEX emissions in these cities.





The 5-month average SBTEX modeled concentrations are close to the average measurement data
(R of total SBTEX is 0.74, Benzene is 0.68, Toluene is 0.45, Xylenes is 0.77, Ethylbenzene is 0.77,
and Styrene is 0.64). These spatiotemporally fine modeled air SBTEX concentrations can be used
for conducting epidemiologic analyses or in risk assessment. The diurnal variation of SBTEX
concentrations that is opposite to its emissions pattern indicates that the concentration is more
sensitive to emission at night than daytime. Therefore, the HAPs emission control policy should
also focus on night-time emissions. Further, the hourly SBTEX data can be used in epidemiologic
analyses to investigate effects of acute exposures and short-term changes in those exposures.
This study acknowledges the considerable uncertainties in this approach, including the accuracy
of emission data, the meteorological condition data, and oxidants concentrations (OH radical, $O_3$,
and $NO_3$) simulation in the CB6 mechanism. There are limited observational data to verify the
model performance. Because the NEI contains bottom-up emission data, all emission rates and
compositions may have bias and be incomplete. Additionally, the emission activity in hourly,
daily, and monthly temporal profile may not fully reflect the actual emission process from large
industry stacks, especially emergency emissions from unreported flare (final treatment equipment)
or leak processes. These detailed emissions are not considered in the NEI. Further, the
concentrations of oxidants are simulated in the CAMx model with the CB6r4 mechanism; this
mechanism is designed to simulate ozone and PM. Therefore, the model species OH radical, $NO_3$,
and $O_3$ may differ from the actual concentrations. These oxidant concentrations affect the chemical
decay rate, especially in big metropolitan cities with higher NOx emissions.
In future work, this study will perform data fusion between these modeled SBTEX concentrations
and the observational data using the Bayesian Maximum Entropy (BME) method to generate a
hybrid concentration map. The BME method will be used to reduce any bias and error of the model
data. The model results can provide estimated SBTEX concentrations in areas lacking monitoring
stations. This will facilitate epidemiologic studies of SBTEX exposures in relation to a range of
health outcomes in the Gulf region and can be extended to provide similar health research
opportunities elsewhere.



**Code availability:**

1. The source code of the CAMx7.00 model and model preprocess tools (O3map, tuv4.8, wrfcamx, camq2camx) can be downloaded on the Environ website: http://www.camx.com (RAMBOLL, 2021)

2. Python 2.7 is used to treat the model output and can be downloaded on anaconda python website: https://www.anaconda.com/distribution/ (Anaconda, 2020)

3. R project for statistical computing can be downloaded at https://www.r-project.org (The R Foundation, 2021)

4. HAPI program code can be downloaded on GitHub: https://github.com/tatawang/HAPI (Wang and Baek, 2023)

**Data availability:**

1. The result of this study, including SBTEX emission, concentration data and evaluation code in this study can be downloaded at: https://zenodo.org/record/7967541 DOI: 10.5281/zenodo.7967541 (Wang et al., 2023)

2. The 2011 NEI emission model platform (EMP) and SMOKE model system can be downloaded on the EPA ftp website: https://www.epa.gov/air-emissions-modeling/2011-version-6-air-emissions-modeling-platforms (USEPA, 2021b)

3. The meteorological data can be found on CMAS Data Warehouse website: https://dataverse.unc.edu/dataverse/cmascenter (UNC-IE, 2021)

4. The AMTIC data can be found at: https://www.epa.gov/amtic/amtic-ambient-monitoring-archive-haps (USEPA, 2021c)

**Author contribution**

CTW and BHB are the lead researchers in this study and are responsible for research design, producing data, experiments, results analysis, and manuscript writing. WV and JX are co-head researchers and guided the research design, assessed model results, and contributed to writing the manuscript. JG, MS, RS, LE, JB, and JHW helped to collect and verify data and write the manuscript.



**Competing interests**
The Authors declare that they have no conflict of interest.
**Acknowledgments**
We want to thank the National Institute of Environmental Health Sciences (NIEHS)
support the research grant (Neurological Effects of Environmental Styrene and BTEX Exposure
in a Gulf of Mexico Cohort, Grant No. R01ES031127); the NOAA Climate Program Office's
Atmospheric Chemistry, Carbon Cycle, and Climate (AC4) program and Climate Observations
and Monitoring (COM) program, Grant No. NA21OAR4310225 (GMU) / NA21OAR310226
(UMD); The Fine Particle Research Initiative in East Asia Considering National Differences
(FRIEND) project through the National Research Foundation of Korea (NRF) funded by the
Ministry of Science and ICT (2020M3G1A1114621); Korea Environment Industry & Technology
Institute (KEITI) through Climate Change R&D Project for New Climate Regime funded by Korea
Ministry of Environment (MOE, Grant NO. 2022003560007). We also thank the Texas
Commission on Environmental Quality (TCEQ), South Coast Air Quality Management District
(AQMD), Ramball CAMx team, and the University of North Carolina at Chapel Hill (UNC-CH)
for their invaluable assistance.



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




## Tables


**Table 1.** The Annual emission rates (metric tons yr$^{-1}$) of Styrene, Benzene, Toluene, Ethylbenzene, and Xylene (SBTEX) in 2012 including the increases resulting from this work. The percent increase from the 2012 National Emission Inventory is given in parentheses. The bold font indicates the emission sector with the maximum SBTEX rates.


| Emission Sectors | BENZENE tons yr$^{-1}$ | TOLUENE tons yr$^{-1}$ | XYLENES tons yr$^{-1}$ | ETHYLBENZENE tons yr$^{-1}$ | STYRENE tons yr$^{-1}$ | Total tons yr$^{-1}$ | Sectoral share of total |
|---|---|---|---|---|---|---|---|
| agriculture fire (agfire) | 1,128 | 745 | 0 | 0 | 0 | 1,873 | 0.76% |
| commercial marine vehicle (cmv) | 103 | 16 | 24 | 10 | 11 | 164 | 0.07% |
| non-point source (nonpt) | 3,070 | 16,932 | 5,156 | 1,188 | 777 | 27,123 | 11% |
| non-road vehicle (nonroad) | 4,752 | 13,506 | 14,265 | 2,682 | 171 | 35,376 | 14% |
| on-road vehicle (onroad) | 10,495 | **43,657** | **27,271** | **7,472** | 309 | **89,204** | 36% |
| wild fire (ptfire) | **46,052** | 10,909 | 4,355 | 0 | 0 | 61,316 | 25% |
| Rail (rail) | 10 | 14 | 20 | 8 | 9 | 61 | 0.02% |
| residential wood combustion (rwc) | 395 | 92 | 26 | 0 | 0 | 513 | 0.21% |
| non-point oil gas industry (np_oilgas) | 5,421 | 2,694 (+69%) | 4,683 (+51%) | 455 (+100%) | 2 (+100%) | 13,255 (+28%) | 5.4% |
| electricity power plants unit (ptegu) | 277 | 131 (+2%) | 60 (+3%) | 35 (+3%) | 7 (0%) | 510 (+1%) | 0.21% |
| point source emission other than electricity generation unit (ptnonipm) | 7,305 | 2,608 (+17%) | 2,644 (+12%) | 667 (+12%) | **2,911** (+34%) | 16,135 (+10%) | 5.9% |
| point source emission of oil and gas industry (pt_oilgas) | 510 | 314 (+25%) | 209 (+24%) | 36 (+24%) | 2 (+100%) | 1071 (+11%) | 0.43% |
| Total | 79,518 | 90,080 (+2%) | 58,713 (+3%) | 12,553 (+3%) | 4,199 (+22%) | 246,601 (+2%) | 100% |








**Table 2.** Normalized Mean Bias (NMB, %) and Correlation Coefficient (R) comparison of average
observational data and model result during the model simulation period, May 1st, 2012 to Sep 30th, 2012
for the 2012 National Emission Inventory (Base), and the emission scenario adjusted in this study (Adj).
Bold font indicates the model improvement, and gray color font indicates poorer model performance.
Also shown is the count (N) of available daily average data across all sites.

|  | Group | N | Benzene | Toluene | | Xylenes | | Ethylbenzene | | Styrene | | SBTEX | |
|---|---|---|---|---|---|---|---|---|---|---|---|---|---|
|  |  |  |  | Base | Adj | Base | Adj | Base | Adj | Base | Adj | Base | Adj |
| **R (daily average comparison for all sites)** | All | 2717 | 0.54 | 0.57 | 0.57 | 0.58 | 0.56 | 0.56 | **0.56** | 0.55 | **0.57** | 0.65 | 0.65 |
| **R (5 month average comparison for all sites)** | All | 46 | 0.68 | 0.46 | 0.46 | 0.79 | 0.77 | 0.76 | **0.77** | 0.63 | **0.64** | 0.75 | 0.75 |
| **NMB (%) (average comparison for all sites)** | All | 46 | 12.53 | -10.2 | **-6.7** | -30.6 | **-24.9** | -25.2 | **-21.4** | 17.4 | 32.1 | -8.8 | **-5.6** |
| **NMB (%) (daily average comparison for all sites)** | Rural | 508 | -22.3 | -10.6 | **-5.4** | -33.2 | **-19.8** | -26.8 | **-23.0** | -63.9 | **-54.8** | -19.3 | **-13.9** |
|  | Airport | 95 | -41.0 | -4.5 | **0.6** | -18.4 | **1.5** | -26.0 | **-19.5** | 34.5 | 42.3 | -11.8 | **-5.0** |
|  | Urban | 272 | 61.7 | 82.9 | 87.5 | -20.9 | **-8.8** | 17.0 | 19.9 | -50.6 | **-39.9** | 32.6 | 39.3 |
|  | Industry | 1842 | 88.0 | -6.6 | **-2.2** | -26.5 | **-23.5** | -9.4 | **-4.9** | 54.6 | 76.1 | 15.5 | 19.0 |









**Figures:**

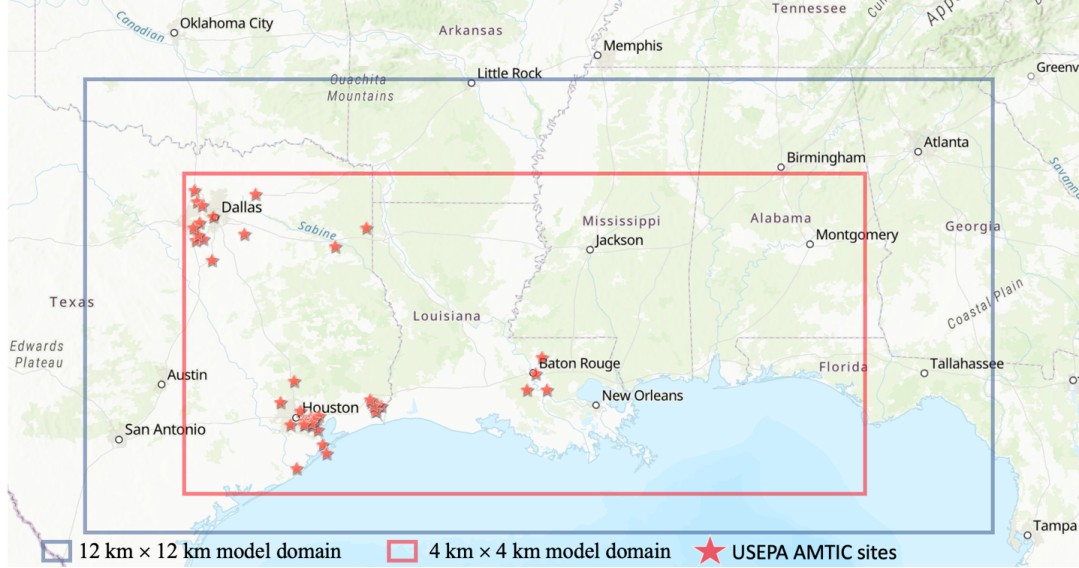

**Figure 1.** The modeling domains with the outer $12 \times 12$ km resolution domains (blue rectangle) and inner $4 \times 4$ km resolution domain (red rectangle). The red stars are the USEPA Ambient Monitoring Technology Information Center observational (AMTIC) sites for Hazardous Air Pollutants (HAPs). There are 4 sites are in Louisiana, and 42 sites in Texas. Generated with ArcGIS map (Esri, 2013).









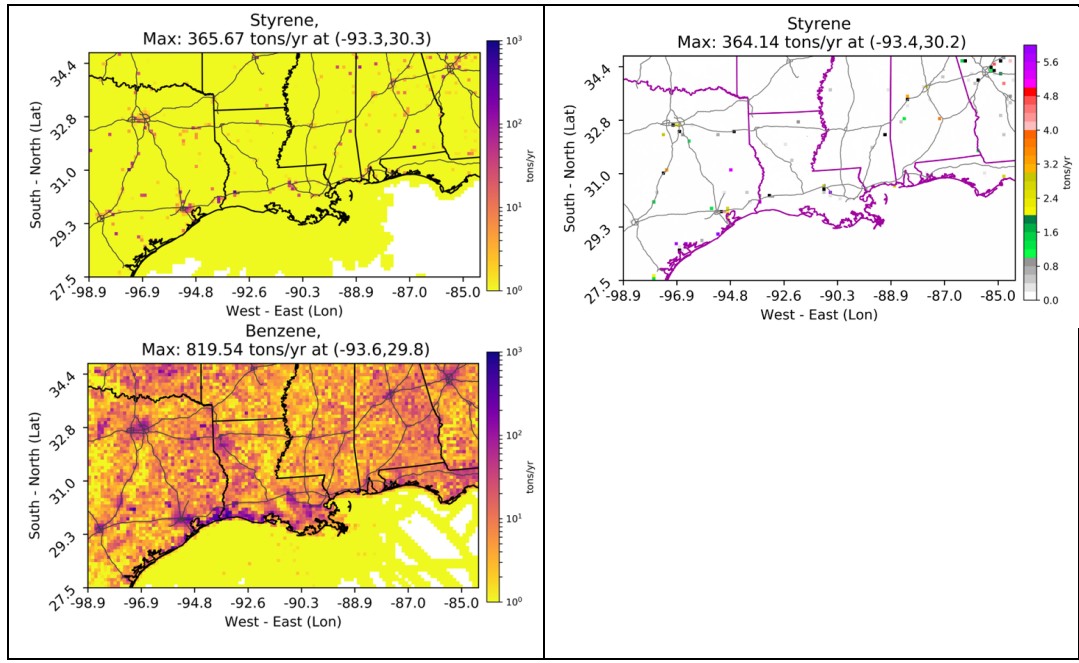

**Figure 2.** Spatial distribution of annual SBTEX emission rates of the modified emission
inventory used in this work, and the location and amount of emissions that were added to the
NEI.




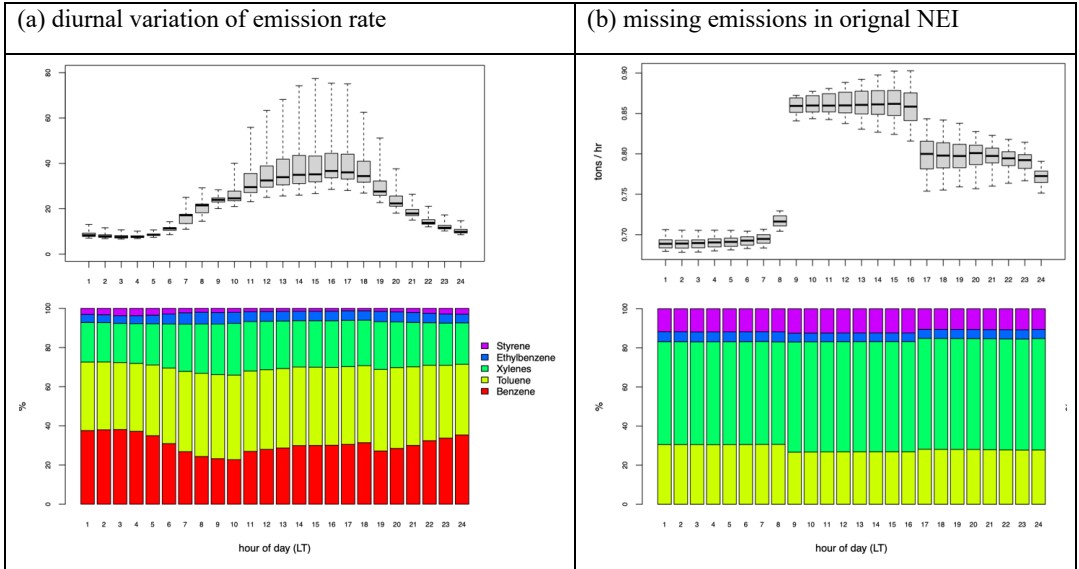


**Figure 3**. Diurnal emission pattern of sum Styrene, Benzene, Toluene, Ethylbenzene, and
Xylenes (SBTEX) (domain total, tons hr$^{-1}$) (upper panel) and the average relative composition of
five species (lower panel).

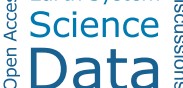





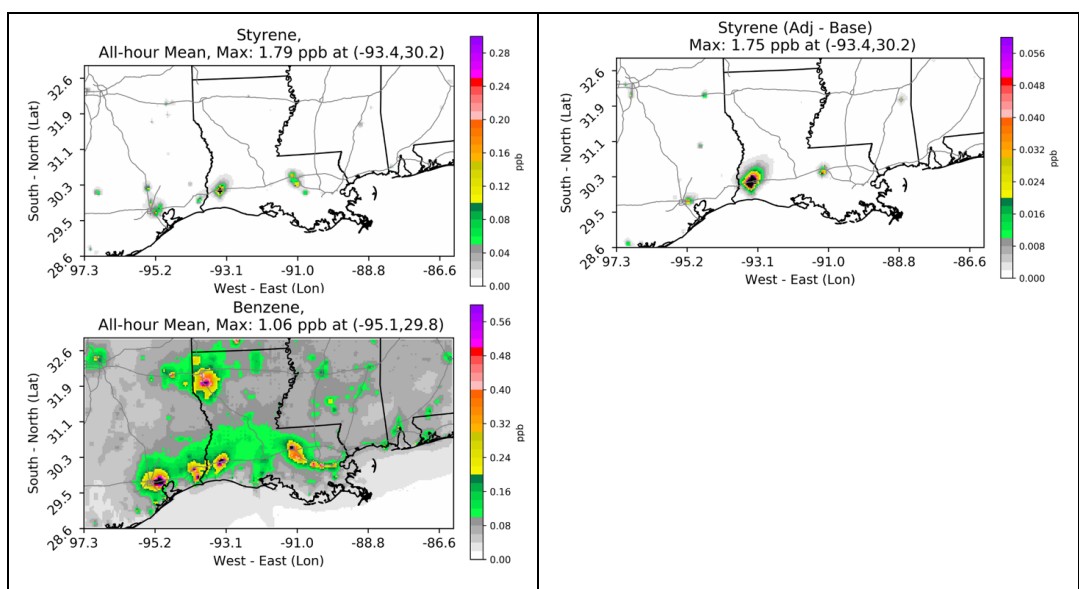

**Figure 4.** The average concentration of SBTEX during the model simulation period (May 1st, 2012 to Sep 30th, 2012) in Adj scenario. The black color indicates the concentration is higher than max color scale bar.

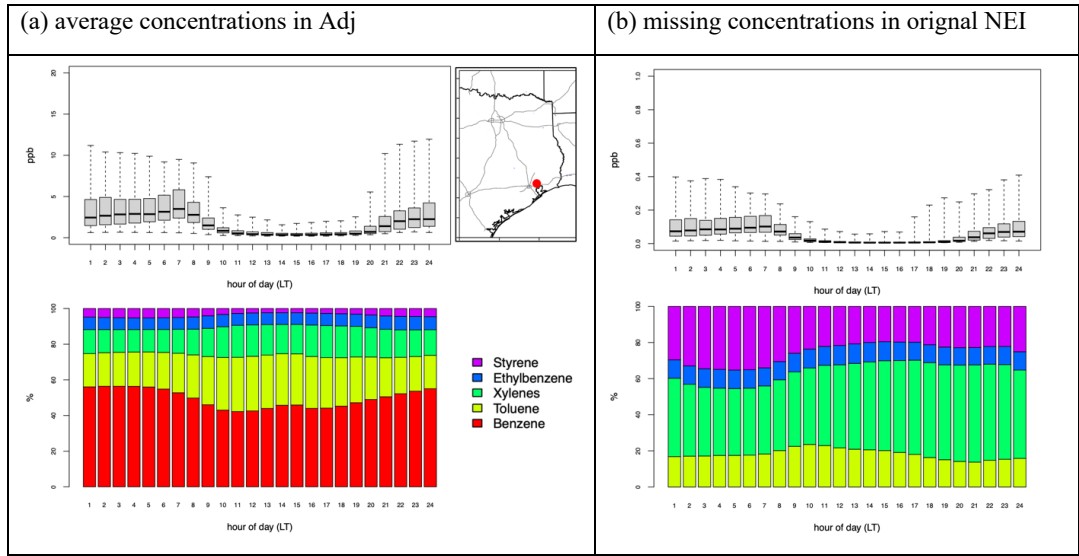


**Figure 5.** Diurnal pattern (upper panel) and relative composition (lower panel) of SBTEX
concentrations from May 1st to September 30th in Houston Ship Channel industry area,
Channelview city (red dot location)

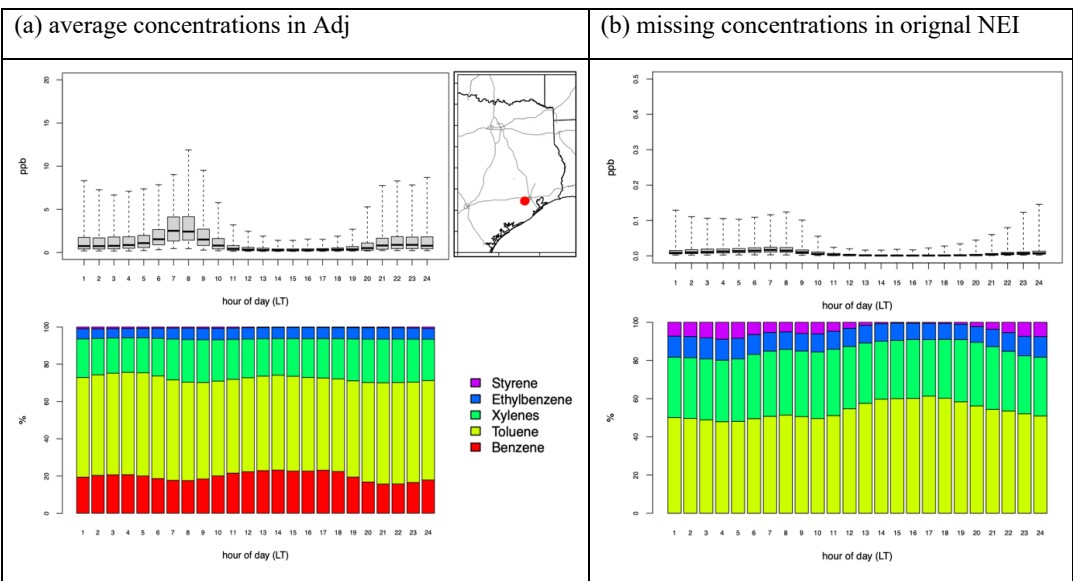


**Figure 6.** Diurnal pattern (upper panel) and relative composition (lower panel) of SBTEX concentrations from May 1st to September 30th in Houston residential area near Bayland Park (red dot location).





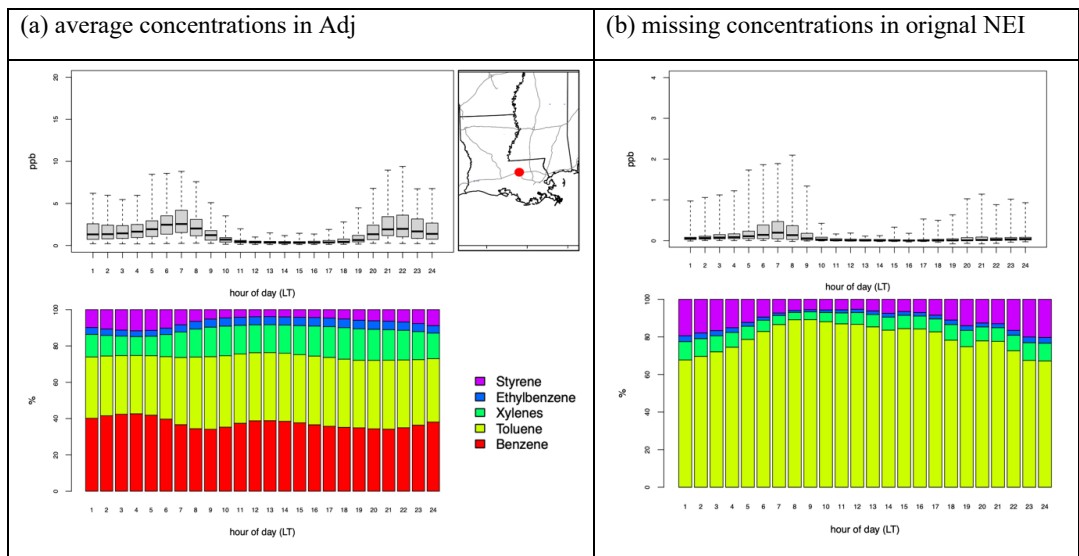


**Figure 7.** Diurnal pattern (upper panel) and relative composition (lower panel) of SBTEX
concentrations from May 1st to September 30th in Baton Rouge city (red dot location).



Data



**Figure 8** (a) The average concentration in Adj scenario overlapped the average observational
measurement data (Diamond shape) during the model simulation period (May 1st, 2012 to Sep
30th, 2012) for (a) Total SBTEX, (b) Benzene, (c) Toluene, (d) Xylenes, (e) Ethylbenzene, (f)
Styrene.

**Earth System** Discussions
**Science**
**Data**

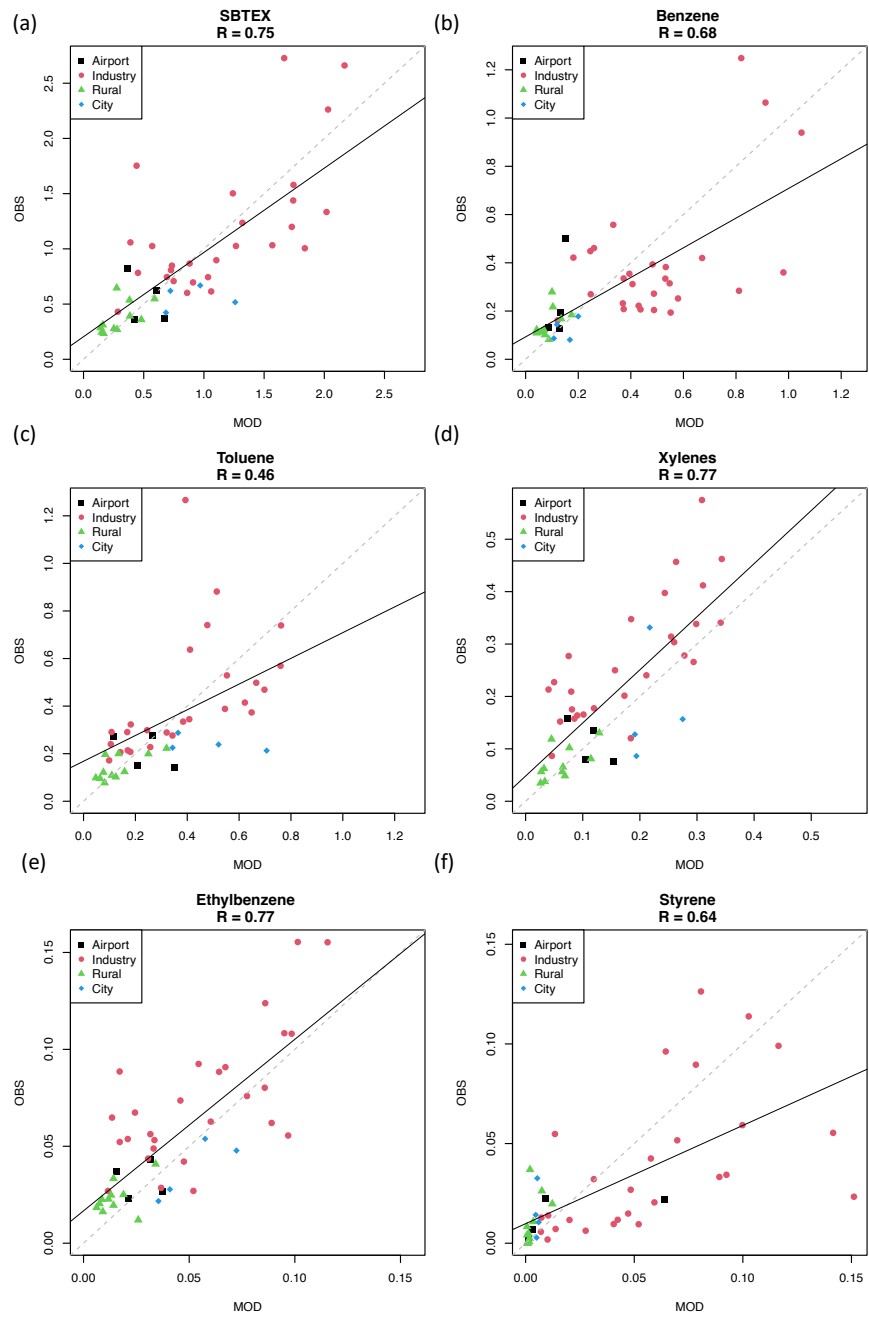

713

**Figure 9.** The average SBTEX concentration (ppb) comparison between model (MOD) and
observational (OBS) data during the model simulation period (May 1st, 2012 to Sep 30th, 2012)
for (a) total SBTEX, (b) Benzene, (c) Toluene, (d) Xylene, (e) Ethylbenzene, and (f) Styrene