# Peer review of "Spatiotemporally resolved emissions and concentrations of Styrene, Benzene,"

_Earth System Science Data, 2023_

## Author Comment (AC1)

**Response to our two anonymous reviewers and one community comment**
Aug 31, 2023

We thank our reviewers for their comments regarding our study. The reviewers' comments are in grey, and authors' responses are given in black.

RC1:
Main Comments:

Improving methods for estimating SBTEX emissions and consequently concentration fields is important for studies on the potential health risks posed by exposure to these pollutants. The HAPI tool developed and released as part of this dataset represents a straightforward and seemingly robust way to make progress on that front by leveraging and combining relevant information currently present in different underlying emission inventories. The other parts of the new dataset, i.e. the new gridded emission files including the imputed SBTEX emissions and the SBTEX concentration fields calculated from these emissions with the CAMx RTRAC tool serve as a nice example of what can be accomplished after applying the HAPI tool. The associated figures and tables documenting features of the gridded datasets for this 5 month case study are well done. That said, given the limited spatial and temporal scope of the gridded emission and concentration fields, the direct use of these aspects of the dataset in future studies may be somewhat limited. Put differently, unless any follow-up studies requiring SBTEX data specifically focus only on this 5-month period in 2012 over this specific area, I expect to see little follow-up use of these portions of the dataset by other groups. The model evaluation R and python scripts used to generate the manuscript figures are nice and easy to follow for anyone familiar with these languages and could easily be adapted to perform similar analyses for other periods or regions.

Thank you for your comments. This study presents a method to identify possible missing STEX emissions in the national emission inventory and simulate the explicit chemical compounds that are not considered in the reduced chemical mechanism. There are very few studies that applied the chemical transport model with the reactive tracer method to simulate SBTEX concentrations for state-level range. Therefore, the same simulation period (May to September 2012) as the Texas Commission on Environmental Quality (TCEQ) State Implementation Plans (SIP) ozone model (TCEQ, 2015) was selected to verify our CAMx core modeling results for the targeted species (O3, NO2, and formaldehyde) that related to oxidants (O3, NO3, OH radical).

Subsequently, this study carefully evaluated our modeling results with USEPA 87 surface ozone monitor sites and 46 AMTIC sites data for May to September 2012. The complete whole 2012 Adj case SBTEX concentration data (January to December) is developed in the final version, along with a monthly evaluation table (Table S9). This 2012 SBTEX concentration data has been uploaded to Zenodo ( https://zenodo.org/record/8303346 ). We have edited the (line267-269 and

line462-464) to explain this. Now it reads:" After we confirmed the model process and SBTEX concentration result was reasonable by observational data, we then conducted this method to generate whole 2012 SBTEX concentration for the U.S. Gulf region and did the model evaluation." and "Beside the 2019 May to September, we also provided the whole 2012 SBTEX hourly concentration data of Adj case in ioapi format, NetCDF format, and comma-separated values (csv). The 2012 monthly model performance table for correlation coefficient and normalized mean bias are in table S9."

This method's configuration and simulation processes are currently applied for additional modeling years from 2011 to 2016 for generating long-term SBTEX concentration data for supporting the National Institute of Environmental Health Sciences (NIEHS) cohort study. The Adj case SBTEX concentration results for 2011 and from 2013 to 2016 will be uploaded and follow the ESSD living data process policy once the QA/QC processes and model evaluation processes are completed.

With a strong background in emissions and air quality modeling over the U.S., the gridded data sets are usable and understandable, but without such a background, I would not consider them to be described and documented in sufficient detail. For example, the zenodo description lists these files as being in netCDF format, but omits the important fact that the spatio-temporal metadata information stored in these files follow I/O API conventions (https://www.cmascenter.org/ioapi/). These conventions, while often used in U.S. emissions and air quality modeling applications, are virtually unknown outside that community, making the interpretation of time and geolocation information of the values stored in these files impossible to many users. At a minimum, the documentation on both zenodo and within the article should make reference to the use of I/O API conventions and link to these conventions, but given the niche nature of I/O API and the learning curve associated with correctly interpreting its geospatial metadata, preferably the time and spatial coordinate and attribute conventions of these files should follow more widely used conventions such as netCDF-CF with explicit latitude, longitude, and time variables included in each file. Likewise, there is no explicit documentation on the meaning of variable names in the emission and concentration files. The "var_desc" attribute could be used to provide such documentation, but it currently just repeats the variable name without providing any further insight into, for example, the differences between "ALD2" and "ALD2_PRIMARY" which are not likely to be known to most readers / users. Documentation should be provided for all variables in all datasets. Because of these shortcomings in metadata and variable documentation, I do not consider the gridded datasets in their current form to be of high enough quality.

Thank you for the comments. We have converted the Adj case file to netCDF-CF format by using Python PseudoNetCDF (https://github.com/barronh/pseudonetcdf) and have uploaded the next version with detailed variable descriptions to Zenodo. For improved readability, we have also

prepared a "comma-separated values (csv)" file containing hourly data for individual SBTEX species in the Gulf Region for 2012. Additionally, we have included "Readme" files with those uploaded data on Zenodo (https://zenodo.org/record/8303346) (Readme_for 2012_SBTEX_conc_4km.txt, and Readme_for_2012_emission_12km.txt).

I am confused by the description of RTRAC as a "post-process feature" / "post-analysis feature", and "post-processing step" in the main article and supplement, and its depiction as a separate box in Figure S1. The RTRAC documentation in the CAMx user guide describes it as a built-in probing tool that, if enabled for a given CAMx simulation, is being applied simultaneously with the base model, and not after the CAMx simulation. Moreover, the description in the text and Figure S1 state that RTRAC tracer concentrations are affected by emissions and physical and chemical decay, but do not mention transport (advection and diffusion). Are the RTRAC SBTEX species not transported from their emission sources? Based on the CAMx user guide, I think they are, but based on the RTRAC method description provided in this article, I was left with the impression that they weren't.

Thank you for correcting this part. Yes, you are right, the RTRAC considers the physical transport processes and it is probing tool. We replace the "post-process" with "probing tool", and correct the RTRAC description in manuscript (line 252-256) to embrace the transportation process. Now it reads "Along with the physical transport processes (diffusion and advection) and decay processes like wet and dry deposition same as core model, there is the second-order chemical reduction rate $r$ that is calculated using the oxidants (Ozone, OH, NO$_3$) concentrations [$Ox$], the SBTEX concentrations [$Tr$], and the rate constants of reactions $k_{Tr+Ox}$ (Eq.3)."

The method description in Section 2.2 of the main article should be improved as it would be critical for any user trying to replicate the CAMx-simulated concentration fields. The details of the air quality model description provided in Section 2 of the supplement should be included in Section 2.2 of the main article (e.g. information on the version of WRF, photolysis rates, and the generation of boundary conditions). There appears to be a contradiction between the main manuscript and the supplement about how WRF fields were prepared for CAMx – the main manuscript states "They were converted to SMOKE- and CAMx-ready gridded hourly meteorology through the Meteorology Chemistry Interface Processor (MCIP)" while the supplement states that "They are converted to the CAMx-ready format using the WRFCAMx version 4.8.1 program developed by the CAMx development team".

Thank you for your valuable comments. We apologize for any confusion, and we have made edits to section 2.2 to provide additional details about the air quality model in section 2.2.1. Regarding

WRFCAMx and MCIP, we have revised line 220, and now it reads: "The WRF output data were transformed into SMOKE-ready gridded hourly meteorology through the Meteorology Chemistry Interface Processor (MCIP). Emissions sectors modulated by meteorology, such as onroad (Choi et al., 2014; Lindhjem et al., 2004) and biogenic, were estimated using the MCIP gridded hourly meteorology. The USEPA's 2012 daily total wildfire emissions (ptfire) estimated by SMARTFIRE2 (USEPA, 2015) were also incorporated (USEPA, 2021). Additionally, the WRF meteorological data were converted to CAMx-ready meteorological data by using WRFCAMx (RAMBOLL, 2020) for the CAMx model input."

The method description should also define "flexi-nesting", a term not likely to be familiar to most readers or users of the dataset. Based on my read, both meteorology and emissions were processed for a 12 km grid, and the 4 km "flexi nest" grid was solely defined during the CAMx simulation with CAMx interpolating 12 km inputs to that finer grid during the simulation. Or were there any inputs actually prepared for the 4 km grid? (the distributed emissions dataset is for 12 km). If none of the inputs were prepared for the 4 km grid, what is the rationale for having CAMx interpolate 12 km inputs to that finer grid?

Thank you for this comment. For the area emission source sectors, there are no additional inputs needed for 4 km × 4 km flexi-nesting domain but the CAMx can interpolate coarse 12 km × 12 km gridded datasets internally. However, the points sources (ptegu, ptnonipm, pt_oilgas, and ptfire) are processed independently with their actual stack locations and their parameters (stack height, outlet temperature, outlet gas speed, and diameter…etc) to enhance their vertical allocations in any modeling domain. Therefore, the CTM model can enhance the horizontal and vertical spatial representations of ambient air pollutants compared to the 12km × 12 km domain. The example figures show a better spatial representation of SBTEX concentrations over the same modeling region between 12 km × 12 km (a), and 4 km × 4 km (b) for 2012.01.01 hour 17 GMT. The flexi-nesting method also allows users to efficiently provide accurate and consistent boundary conditions of ambient air pollutant concentrations for the 4km x 4km modeling simulations. We have edited the line 233-235 and clarified this information. Now it reads "The point source emissions are processed independently with their stack locations in the model domain and considering the plume-raising effect by stack parameters. As a result, the model spatial allocations can be enhanced through the flexi-nesting method."

[Figure]

I also cannot reconcile the fact that CAMx RTRAC simulations apparently were performed for that 4 km grid (implying that having 4 km concentration fields is desirable) with the fact that for model evaluation, the 4 km grid results are aggregated back up to 12 km by considering all nine grid cells surrounding a monitor (implying that modeled gradients at 4 km are not expected to be realistic).

Thank you for the comments. the evaluation method is based on USEPA's evaluation process (USEPA, 2006), the 4 km × 4 km consider the monitoring sites located within the grid cell with surrounding 8 grid cells; therefore, these 9 grid cells average are not the same as 12 km × 12 km domain.

The structure and language of the article and supplement are generally good, but there are a number of instances with somewhat awkward wording, with a few examples noted in the specific comments below. During revision, the article would benefit from a careful editorial review.

Thank you for your comments. We followed your specific comments listed and have edited below:

Specific Comments:

Line 49: Here and in all subsequent references to (USEPA, [year]) in the main article and supplement: please double check that all EPA citations listed in the references section actually have a publication year associated with them. At first glance, only the "Guidance on the Use of models and other analyses for demonstrating attainment of air quality goals for ozone, PM2.5 and Regional Haze" [which was actually updated in 2018] and "2014 Fire NEI Workshop Emissions

Processing-SmartFire Details" references currently do, and yet there are many "(USEPA, [year])" citations throughout the text that don't have a clear connection to an item in the reference list.

Thank you for the comments. We corrected the issues in the reference list and citations.

Lines 108 – 110: see main comments above, is the reactive tracer function really a "post-process"?

Thank you, we have edited the reactive tracer in method section. it reads (line 108-110) "Because the current CAMx model simulation process cannot support explicit SBTEX simulation with reduced chemical mechanism, one of the CAMx probing tools called reactive tracer function is used to overcome the limit of the reduced mechanism"

Lines 136 – 140: The first sentence says emissions "from certain facilities" are collected by EIS and then used to develop the NEI while the last sentence then says the NEI has "emissions … from all types of emission sources". Please clarify

This paragraph explains that USEPA NEI data can have both Hazardous Air Pollutants (HAPs) and Criteria Air Pollutants (CAPs) emission data. However, the HAPs data is voluntary reported by state agencies. We removed the unclear description that could confuse the readers, and now it reads (line 136-142):"The NEI is a national database providing comprehensive annual air emission estimates for both criteria air pollutants (CAPs) (e.g., CO, $NO_X$, $SO_2$, $NH_3$, VOC, and $PM_{2.5}$), and HAPs (e.g., benzene, acetaldehyde, formaldehyde, xylenes, styrene, and more) from all types of emissions sources (e.g., point, nonpoint, and mobile). While CAPs emissions are reported by the state agencies is mandatory, the report of HAPs is voluntary. Consequently, only the limited set of HAPs have been reported to the USEPA, and their spatial coverage can vary significantly by source type (e.g., industrial, vehicles) and region (e.g., county and state) (Strum et al., 2017). "

Thank you, we have corrected the typo.

Line 145 – 146: Does this distinction indeed apply to the NEI, or more broadly to an emissions modeling platform that includes the NEI but also tools like SMOKE and speciation databases? Specifically, does the NEI actually include "model surrogate" species, or are they only computed

from NEI information (VOC and/or HAP explicit) during emissions processing for a specific chemical mechanism in a specific modeling platform?

Thank you for correcting this part, the NEI can have CAPs and HAPs, but doesn't have model surrogate species, which is generated by VOC speciation profiles during the chemical speciation processing step in the SMOKE modeling. We edited this description, and now it read (line 143-148): The VOC emission species generated by SMOKE from NEI have three types, "model surrogate", "model explicit", and "HAPs explicit" species. The "model surrogate species", such as XYL (Xylene and other poly-alkyl aromatics), TOL (Toluene and other mono-alkyl aromatics), and PAR (paraffin carbon bond), are calculated by VOC speciation profiles in emission model platform and used to predict ozone and secondary organic aerosol (SOA) in the CTM but not for individual HAPs emission and simulation.

Line 193: should "Evoc" be "Evoc,s,f"?

Thank you, it has been corrected. Now it reads (line 195) :"$E_{voc,s,f}$ is the CAP VOC emission for the SCC in the county."

Lines 198 – 199: unclear writing. Perhaps "Only the emission sources for which the sum of all HAPs is zero (sum Eisf = 0) are considered as the "without HAPs" group" instead?

Thank you, we have edited this (Line 200). Now it read "Only the emission sources for which the sum of all HAPs is zero ($\sum_i E_{i,s,f} = 0$) are considered as the "without HAPs" group."

Line 218: "The emissions sectors modulated by meteorology" instead of "meteorology-induced"?

Thank you, we have edited this sentence. Now it read (line 221) "The emissions sectors modulated by meteorology, such as onroad (Choi et al., 2014; Lindhjem et al., 2004) and biogenic, are estimated with the MCIP gridded hourly meteorology."

Line 221: "used" rather than "imported"?

Thank you for your edit. We have edited this sentence and now it read (line 223-224): "The USEPA's 2012 daily total wildfire emissions (ptfire) estimated by SMARTFIRE2 (USEPA, 2015) were also incorporated (USEPA, 2021)."

Lines 224 – 230: See main comment above regarding clarification of RTRAC.

Thank you, we incorporated your comments and have edited all RTRAC description part.

Line 245: please define "long-term" and justify why the focus was on long-term evaluation.

To accurately reflect the time period, we replaced "long-term" with "5 months (May to September 2012)".

Lines 252 – 254: this suggests that only high outliers (2*IQR above Q3) were removed while no screening was performed for low outliers. If so, please state explicitly and justify.

The USEPA has already conducted QA/QC for the AMTIC data. As a result, all values from AMTIC are considered genuine data. The presence of high outliers is attributed to fugitive emission events or random releases of VOC from the oil and gas industries (Couzo et al., 2012). These events are challenging to represent accurately using regulatory emission models and air quality models. However, these exceptionally high data points have a notable impact on the model's performance, including correlation coefficients and mean bias. Consequently, only the high outliers (random VOC emission events) have been eliminated. We have addressed this in the manuscript (line 280), which now reads (lines 283-288): "The USEPA conducted QA/QC for the AMTIC data, which contain values that are exceptionally high due to unpredictable industrial VOC release events (Couzo et al., 2012). These VOC emission events cannot be accounted for in the NEI and are not adequately represented by regulatory emission data and air quality models, particularly in petrochemical, oil, and gas industrial areas. As a result, this study removed outliers (those beyond twice the interquartile range (2×IQR) above Q3) to enhance the model's performance."

Line 265: I believe this should be "vessel", not "vehicle"

Thank you for correcting this in the manuscript. Now it reads (line 300): "commercial marine vessel (cmv)"

Thank you for correcting this in the manuscript. Now it reads (line 325): "some rural areas in Texas"

Thank you, we have edited this in the manuscript. Now it reads (line 336)

Lines 318 – 324: There are some awkward sentences in this paragraph, please review and revise.

We have edited the manuscript (line 403-411). The revised version is as follows: "The spatial distribution patterns of individual SBTEX compounds exhibit similarities due to shared emission sources, except for Styrene. Styrene primarily originates from ptnonipm, while other species predominantly arise from vehicle emissions and wildfires. Benzene (max: 1.06 ppb), Toluene (max: 1.01 ppb), and Ethylbenzene (max: 0.16 ppb) reach their highest concentrations in Houston, reflecting their significant emissions. Further, Xylenes (0.78 ppb) originate from sources in Shreveport. Remarkably elevated concentrations of Styrene (reaching 1.97 ppb) are conspicuously identified proximal to Lake Charles, a locale characterized by an abundant emission of styrene from non-Electricity Generating Unit point sources, which have been absent in the original NEI records."

Lines 342 – 345: The first line states that it's mostly meteorology, but then the third line says both meteorology and more active daytime chemistry and the associated chemical loss of these tracers act to decrease daytime tracer concentrations. Review and revise.

We edited the manuscript (line 429-434). Now it reads: "In general, the diurnal variations of SBTEX concentrations are primarily influenced by various factors (such as ventilation, emissions, diffusion, deposition, and chemical reactions). These variations typically manifest with lower concentrations during the daytime compared to nighttime due to increased ventilation, diffusion, and chemical loss, even though emissions are about 4 times higher during the daytime, as presented earlier (Fig. 4)."

Line 346: "sensitivity of the concentration to emissions" instead?

We updated the manuscript (line 430). Now it reads: "Diurnal meteorological and emission patterns suggest more sensitivity of the concentrations to the emissions during nighttime than daytime, implying that implementing emission controls to reduce the concentrations at night would be most effective."

Line 347: Are nighttime exposures when people are mostly at home a concern, and should lowering concentrations during that time period be a priority?

The higher concentration period should be given priority. However, this statement also depends on the locations (residential, industrial, or urban areas) and human activities. We believe that individuals with more nighttime activities, such as night workers in industrial areas or airport employees working at night, may be more susceptible to the effects of these toxicants in urban environments. Furthermore, houses lacking proper insulation systems or inhabited by low-income populations do not use air conditioning, which could directly expose to these high-concentration toxicants. This study offers a comprehensive SBTEX concentration map for the Gulf region to identify hotspots, and its findings can be applied in future epidemiological studies on human exposure. To address the review concern. We addressed in MS (line 489-491), now it reads" The high SBTEX concentration during nighttime affects individuals who engage in more nighttime activities or reside in houses lacking isolation of outdoor air."

Line 356: Suggest not starting this sentence with "in contrast" because it actually says that the level of peak SBTEX is the same for both locations. Instead, start the next sentence "In contrast to Channelview, the peak concentration at Bayland Park occurs …."

Thank you for this suggestion; we edited this part (line 444-449). Now it reads "The second case, Bayland Park (Latitude: 29.69, Longitude: -95.49), located nearby at the western side of Houston, presents the same level of peak SBTEX concentration (about 12 ppb) (Fig.9a) as Channelview city. In contrast to Channelview, the peak concentration of Bayland Park occurs at traffic rush hour (LT 7:00 to 8:00), contributed mostly by Toluene (53%) and Xylene (23%) (indicating the mobile vehicle sources) rather than Benzene (18%)."

Line 361 – 362: awkward wording of the first part of the sentence, review and revise.

Thank you, we edited this in the manuscript (line 449-451). Now it reads "Meanwhile, the adjusted industry emission sources, as presented in table S5, play a significant role in driving the peak concentration (0.4 ppb) in Channelview city (Fig. 8b), yet exhibit a reduced impact on Bayland Park (Fig.9b), where is far from the industry area."

Line 371 – 372: change "composition of" to "contribution from"?

Thank you, we edited this in the manuscript. Now it reads (line 460): "The missing emission sources (Fig. 10b) will further enhance the peak concentration by 2 ppb at LT 5:00 - 8:00, with the largest chemical contribution from Toluene (about 70 - 85%), followed by Styrene (about 7 - 20%) associated with the industrial sources."

Line 420: "applied" rather than "implemented" since no modifications to off-the-shelf CAMx code were made?

Thank you, we edited this in the manuscript. Now it reads (line 469) : "Then we successfully applied the state-of-the-science chemical transport modelling system……"

Line 429: "underestimated" instead of "missed"?

Thank you, we edited this in the manuscript. Now it reads (line 479): "……but were substantially underestimated in the original NEI data……"

Line 493 – 495: awkward wording, review and revise.

Thank you, we edited the manuscript (line 548-551). Now it read: We want to thank the National Institute of Environmental Health Sciences (NIEHS) support this study, the research project is "Neurological Effects of Environmental Styrene and BTEX Exposure in a Gulf of Mexico Cohort" (Grant No. R01ES031127). We also appreciate the grant support including……

Line 504: Ramboll, not Ramball.

Thank you for correcting this typo in manuscript (line 560).

RC2

This study uses observations and modeling to create an emission map of five hazardous air pollutants (HAPs) for May-Sep 2012 in the U.S. Gulf region. This map includes emissions that are missing in the US EPA emission inventory. High emissions were reported in the literature for these five HAPs in the Gulf region and they pose health risks, therefore an accurate emission inventory is beneficial. The analysis methodology is sound. However, I have some concerns listed below.

Major comments:

The authors should put more effort into justifying the use of various datasets. The current time coverage of the emission map generated is only five months in 2012. Why do authors not use more up-to-date datasets to generate an emission map for longer and more recent time periods? The AMTIC ambient measurements are available until 2023, and US EPA NEI has the most recent version for SBTEX emissions in 2023. So I'm confused why the authors used 2012 ambient measurements and 2011 NEI. If authors find 2012 emissions very interesting, I encourage them to extend the method to the most recent (at least until 2022) to have a 10-year monthly emission map for the Gulf region. This will make the study more interesting to the community.

Thank you for your comments. Thank you for your feedback. There are very few studies that have applied reactive tracers to simulate SBTEX concentrations. Therefore, this study meticulously verified the model processes. We selected the same simulation period (May to September 2012) as the Texas Commission on Environmental Quality (TCEQ) State Implementation Plans (SIP) ozone model (TCEQ, 2015) to validate our CAMx core modeling results. We accomplished this by comparing concentrations of ozone, NO2, and formaldehyde. Those species are connected to oxidants (O3, NO3, and OH radicals) in the atmosphere and contribute to the chemical decay of SBTEX. Additionally, the core model results were evaluated for ozone performance using USEPA surface air quality stations. Furthermore, the SBTEX concentrations derived from the reactive tracer process were compared with observational data from AMTIC.

Moreover, the emission data used to generate the concentration map were sourced from NEI, with the latest available data year being 2019 (released in Sep. 2022, https://www.epa.gov/air-emissions-modeling/2017-2019-air-emissions-modeling-platforms). Therefore, we are unable to update the SBTEX concentration map to 2022 now. However, this study is in collaboration with the National Institute of Environmental Health Sciences (NIEHS) SBTEX exposure cohort study for the Gulf region for 2011-2016. We have uploaded the 2012 Adj case concentration data ( https://zenodo.org/record/8303346 ), confirming its monthly performance (correlation coefficient and normalized mean bias) in table S9. The Adj case SBTEX concentration results for the other years (2011, 2013-2016) will be uploaded and will follow the ESSD living data process policy once their QA/QC processes and model evaluation are completed.

Line 96-97: why dispersion models cannot support regional scale application? Also add references.

The dispersion models are employed to simulate air pollutant concentration at the downwind location (USEPA, 2023). The dispersion models are driven by an hourly single wind field (please see "https://gaftp.epa.gov/Air/aqmg/SCRAM/models/preferred/aermod/aermod_userguide.pdf" page 3-134 surface meteorological data input). The dispersion model considers only a single wind field and very simple chemistry process for all receptors (USEPA, 2022). In contrast to dispersion model, chemical transport models (CTM) are the Eulerian models, which consider multiple grid cells (3 dimensions and time), and each grid cell has its own meteorological input. Consequently, the CTM model can simulate air quality on state-level scale with the chemistry processes. We added the references for this description and have edited the "regional scale" to "state-level scale." Now it reads (line 96-98): "These dispersion models, however, account for exposures at a small spatial scale due to its simple meteorology process (USEPA, 2022, 2023) and cannot support state level scale application."

Line 129: provide reference for NEI 2011.

Thank you, we added the reference to the manuscript.

Line 130-133: explain the difference between NEI vs. TRI vs. EIS.

NEI (National Emission Inventory) is the emission database that considers all anthropogenic emissions including criteria air pollutants (CAPs) (e.g., CO, $NO_X$, $SO_2$, $NH_3$, VOC, and $PM_{2.5}$) and hazardous air pollutants (HAPs) (e.g., Benzene, Acetaldehyde, Formaldehyde, Xylenes, Styrene, and more). TRI (Toxics Release Inventory) is the toxics (or HAPs) emission inventory but only for special emission sources especially the larger HAPs emission sources. EIS (Emission inventory System) is the interface for state agencies to report the emission data including CAPs and HAPs to USEPA. We edited this paragraph for clarification. Now it reads (line 128-133) "The emission inventory used as a base was the 2011 version 6 NEI (USEPA, 2021). Subsequently, the SMOKE modeling system was employed to produce hourly gridded emissions of SBTEX across the Gulf modeling region for the year 2012. These SBTEX emissions were utilized in conjunction with the CTM model and a reactive tracer function to generate the SBTEX concentration map. In the end, the USEPA Ambient Monitoring Technology Information Center (AMTIC) data were employed to evaluate our model's outcomes.

Line 140: lowercase for "Styrene"

Thank you for correcting this in the manuscript. For consistency in the manuscript, we have changed all chemical compounds full name to lowercase.

Line 197-198: this means that their emissions are assumed to be correct? Having no missing emission doesn't mean it is correct. Uncertainty and evaluation against observations should also be conducted for them.

This study assumed that when VOC and their associated HAPs emissions are present in the NEI, we consider them to be consistently estimated. It's accurate to acknowledge that this doesn't necessarily imply their correctness, but our assumption is that HAPs have been reported alongside the VOC emissions. For instance, using SCC code "2310021010," which corresponds to "Industrial Processes; Oil and Gas Exploration and Production; On-Shore Gas Production; Storage Tanks: Condensate," we assumed that the same emission processes will result in similar VOC and HAPs compositions. Hence, when NEI HAPs data share the same SCC code "2310021010" but display all HAPs as zero, we interpret this as missing HAPs emissions. We are aware of the uncertainties inherent in this approach; therefore, in section 3.3, we compared the CTM model's results with observational data to assess these uncertainties. Furthermore, we have discussed these uncertainties in our conclusion and discussion.

Section 2.2: what is the spatial and temporal resolution of CAMx RTRAC?

The spatial resolution is 4 km × 4 km, and the temporal resolution is hourly. We edited sector 2.2 Model configuration part to show more model setup details in the manuscript.

Line 218-220: should include more details about meteorology-induced emissions sectors, biogenic and wildfire emissions in the main manuscript, I see they are now in SI.

We have edited sector 2.2 Model configuration part and add more basic model details to this sector.

Figure S1: I recommend including it in the main manuscript.

Thank you for this comment. We have added this figure as figure 2 in sector 2.2 in manuscript.

Section 3: should provide more specifics on the confidence intervals, and uncertainty range of the emission rates or concentration reported.

We acknowledge the significance of data uncertainty. Regarding the uncertainty related to emission data, it's unfortunate that the USEPA has only provided an annual emission database and emission model platforms. There are no report available detailing emission uncertainty ranges that have been estimated by the USEPA and published in NEI or incorporated into the emission model platforms. Certain studies have employed their own methods to estimate potential emission uncertainties (Frey, 2007; Yan et al., 2014; Zhao et al., 2017). However, this aspect falls outside the scope of our current research. Regarding the concentration part, which is the focus of our study, section 3.2 of the results presents the model evaluation for SBTEX using correlation coefficients and normalized mean bias to illustrate the model's performance. These evaluation methods have been developed by the USEPA (USEPA, 2006) and the air quality model community.

Figure 2, 3: should include time periods in the figure legend.

We edited the Figure caption (now Figure 3) with more details for the time periods. Now it reads "Spatial distribution of 2012 annual total SBTEX emission rates (ton/yr) of the modified emission inventory used in this work (a), and the location and amount of emissions that were added to the NEI (b)."

Section 3.3: as model validation should come as section 3.1.

Thank you for the comment. We put model result evaluation into section 3.2 (previous version is 3.3), and SBTEX concentration pattern description into 3.3 (the previous version is 3.2).

Table 2: (1) should provide such information for each site, each month and individual pollutant to show how they differ. (2) it seems that including missing emissions ("Adj") does not improve R values, any reason for that?

(1) Due to the fact that many sites only provide weekly data (with one day average per week), we have incorporated a monthly model evaluation table for correlation coefficient and normalized mean bias across all sites in tables S9a and b.

(2) The model outcomes are the result of multiple physical and chemical processes, encompassing meteorology, chemistry, emissions, and deposition. The principal factors contributing to daily variations are driven by emissions, chemical decay, and meteorological conditions. In this study, we only adjusted the annual total STEX emission. The annual STEX emission (measured in ton yr$^{-1}$) within NEI is processed by SMOKE to generate gridded hourly data utilizing monthly, weekly, daily, and hourly temporal profiles. These temporal profile divisions are the same as the Base case (for example, the weekday rush hour or reduced weekend traffic emissions in urban areas). The R values are primarily influenced by concentrations ranging from low to high. Consequently, there is not a significant enhancement in R-values within the Adj case. We have edited the conclusion and discussion part for addressing this in manuscript. Now it read (line 497-503): " Due to the fact that the NEI is constructed using bottom-up emission data, there is the potential for emission rates and compositions to display bias and incompleteness, all emission rates and compositions could exhibit bias and incompleteness. Despite our implementation of imputation for the HAPs annual data, the emission activity within hourly, daily, and monthly temporal profiles remains unchanged, potentially failing to accurately represent the overall emission process. Additionally, the broader emission processes, including emergency emissions from unreported flaring (such as final treatment equipment) or leakage events, have not been considered in the NEI."

Figure S8: there is a > 2-factor difference between observation and model for BTES from 10 pm to 8 am, this should be discussed.

Thank you. We added a paragraph to discuss the details in the manuscript. Now it reads (line 387-397): "This study reported the diurnal variation of SBTEX concentrations for Houston (using data from only three monitoring sites within the city) in Fig. S8. The hourly data revealed that Benzene, Ethylbenzene, and Styrene are overestimated during nighttime hours (from LT 20:00 to 6:00). Toluene is underestimated at nighttime, whereas Xylene closely aligns with the observed data range in Houston. Except for Benzene, all species experience underestimation during daytime hours (from LT 11:00 to 17:00). The modeled hourly concentration during nighttime can be influenced by chemical reduction and uncertainties in hourly emission patterns. Due to the low planetary boundary layer height and lower chemical reduction processes during nighttime, emissions are more sensitive to concentration changes. The model's diurnal patterns suggest that the hourly emission rate may overestimate during nighttime but underestimate during the daytime in Houston."

Figure 8: color of diamonds are observational data? It is hard to compare observation with model results in this figure.

Thank you for providing these comments. We have revised this figure (now labeled as Figure 5) by using larger diamond shapes and zooming in on the details. The diamond colors correspond to the observational results, and they are intended to closely match their background color (representing model results). This alignment signifies that the model results closely match the observational data and capture concentration gradients. Additionally, Figure 6 illustrates a one-to-one data comparison of the data presented in Figure 5 for all SBTEX species.

Section 4: should compare the new emission map to NEI and discuss how US EPA can improve their estimates, potentially also provide a correction factor for NEI to match the new emission map.

Figure 3b illustrates the difference for STEX emission map, and table S5 display the emission difference by states and emission sectors. Furthermore, both table S6 and table S7 provide the detailed breakdown of the potential missing emissions by SCC. The result shows that the non-point oil and gas sector (np_oilgas) have largest difference, and point source emission other than electricity generation unit (ptnonipm) have the second largest possible missing emission. Detailed comparisons between the NEI and new HAPs emission data are available in Supplementary Document Section 3, titled "Missing HAPs Emission from oilgas and ptnonipm. In the correction factor aspect, we did not generate a correction factor specifically for the NEI. Instead, we employed the HAPI program to determine the ratio of HAPs to VOC emission data by NEI, enabling us to generate SCC-level HAPs to VOC ratios. Subsequently, we utilized this ratio to perform the imputation process.

References: some references missed the publication year or the year of last access date.

Thank you for correcting this, we have edited the revision and fixed this issue.

In Supplementary material: line 67: hydroperoxyl radical is HO2, OH is hydroxyl radical

Thank you for correcting this, we fixed this issue.

CC1

The paper's benefit is demonstrating a method to account for HAP emissions missing in the US EPA NEI. The method determines these emissions by linking sources with reported VOC emissions to their missing HAP emissions. The link is the same type of source that the NEI gives both VOC and HAP emissions. The demonstration includes eulerian model (*a version of CAMx*) simulations that use the origin and corrected emission inventories for five HAPs, *benzene, toluene, ethyl benzene, styrene, and xylene isomers*. The eulerian model is not a full photochemical model to tropospheric chemistry but a tracer model with transport, deposition and simple chemical destruction. The chemical destruction uses oxidant concentrations predicted from previous photochemical model (*a different version of CAMx*) simulation. The method seems reasonable, and the demonstration seems to improve model predictions compared against observations. The paper does not conclude whether the method can be use for other HAPs such as polycyclic aromatic hydrocarbons. Perhaps, the paper's authors can update the conclusion section with this information.

This method can be applied to other explicit HAPs and can incorporate both chemical and physical processes. We have considered the reviewer's suggestion and included it in the conclusion section. The revised statement now reads, "In future studies, this approach can be extended to other chemical compounds to estimate their concentrations. The USEPA provides emission data for approximately one hundred HAPs in the NEI for certain emission sources. Those emission data can also be processed to derive HAPs concentrations."

This reviewer has a few minor problems with the paper. The main text puts details on the photochemical modeling and its evaluation into supplementary materials. In the main text, Model Configuration section could summarize them such as the chemical mechanism and how oxidant observations compare model predictions such as for ozone. The same section should also state whether the photochemical and tracer simulation used the same meteorological data.

Thank you for this comment. We added more model details in section 2.2 and add the model evaluation by AQS data and TCEQ SIP model result. And yes, the RTRAC method applied the same meteorological data.

The point is only implied. Also, the paper does not state if tracer simulations uses boundary and initial conditions for any of the five HAPs.

The boundary and initial conditions do not have any SBTEX. To reduce the boundary condition effect, we applied the $12 \times 12$ km domain to process the $4 \times 4$ km nest domain (Figure 1). All model result present in this paper is $4 \times 4$ km only. For the initial condition part, we processed the

model spin-up on April 20th ,2012, so there is 10 days before the model simulation period (May 1st to Sep. 30th, 2012).

Finally, the reviewer wonders how benzene predictions compare between the photochemical and tracer simulations because the cb6r4 mechanism has a species representing benzene called BENZ at least according to the CAMx User Guide.

We compared the photochemical and tracer BENZ result, their concentrations across all model domain are very close; their concentration difference are less than 0.1%.

Overall, this reviewer believes that the paper is worthy of publication with possible revisions as suggested above.

Thank you.

**Reference list:**
Choi, K.-C., Lee, J.-J., Bae, C. H., Kim, C.-H., Kim, S., Chang, L.-S., Ban, S.-J., Lee, S.-J., Kim, J., and Woo, J.-H.: Assessment of transboundary ozone contribution toward South Korea using multiple source–receptor modeling techniques, Atmospheric Environment, 92, 118-129, https://doi.org/10.1016/j.atmosenv.2014.03.055, 2014.

Couzo, E., Olatosi, A., Jeffries, H. E., and Vizuete, W.: Assessment of a regulatory model's performance relative to large spatial heterogeneity in observed ozone in Houston, Texas, Journal of the Air & Waste Management Association, 62, 696-706, 10.1080/10962247.2012.667050, 2012.

Lindhjem, C., Chan, L., Pollack, A., Corporation, E. I., Way, R., and Kite, C.: Applying Humidity and Temperature Corrections to On and Off-Road Mobile Source Emissions, 13th International Emission Inventory Conference, Clearwater, FL2004.

RAMBOLL: CAMx7.00 User's Guide: https://camx-wp.azurewebsites.net/Files/CAMxUsersGuide_v7.00.pdf, last 2020.

Strum, M., Kosusko, M., Shah, T., and Ramboll: SPECIATE and using the Speciation Tool to prepare VOC and PM chemical speciation profiles for air quality modeling,

https://cfpub.epa.gov/si/si_public_file_download.cfm?p_download_id=532731&Lab=NRMRL, 2017.

USEPA: Guidance on the Use of models and other analyses for demonstrating attainment of air quality goals for ozone, PM2.5 and Regional Haze, https://permanent.fdlp.gov/LPS74329/draft_final-pm-O3-RH.pdf, 2006.

USEPA: 2014 Fire NEI Workshop Emissions Processing-SmartFire Details, https://www.epa.gov/sites/default/files/2015-09/documents/emissions_processing_sf2.pdf, 2015.

USEPA: 2011 Version 6 Air Emissions Modeling Platforms: https://www.epa.gov/air-emissions-modeling/2011-version-6-air-emissions-modeling-platforms, last access: April, 04, 2022, 2021.

USEPA: User's Guide for the AMS/EPA Regulatory Model (AERMOD): https://gaftp.epa.gov/Air/aqmg/SCRAM/models/preferred/aermod/aermod_userguide.pdf, last access: August 15, 2023, 2022.

USEPA: Air Quality Dispersion Modeling: https://www.epa.gov/scram/air-quality-dispersion-modeling#:~:text=Dispersion%20modeling%20uses%20mathematical%20formulations,at%20selected%20downwind%20receptor%20locations., last access: August 15, 2023, 2023.

---

## Author Response (AR1)

**Response to our two anonymous reviewers**
Sep 30, 2023

We thank our reviewers for their comments regarding our study. We have modified the manuscript and provided a point-by-point response since the last version. To streamline the process and avoid confusion, we have decided to include the data description for May to September 2012 in this manuscript. The future updated data, encompassing the years 2011 to 2016 and additional information on model performance table, will be provided under ESSD living data policy.
Within this point-to-point response document, the reviewers' comments are highlighted in grey, while the authors' responses are presented in black.

RC1:
Main Comments:

Improving methods for estimating SBTEX emissions and consequently concentration fields is important for studies on the potential health risks posed by exposure to these pollutants. The HAPI tool developed and released as part of this dataset represents a straightforward and seemingly robust way to make progress on that front by leveraging and combining relevant information currently present in different underlying emission inventories. The other parts of the new dataset, i.e. the new gridded emission files including the imputed SBTEX emissions and the SBTEX concentration fields calculated from these emissions with the CAMx RTRAC tool serve as a nice example of what can be accomplished after applying the HAPI tool. The associated figures and tables documenting features of the gridded datasets for this 5 month case study are well done. That said, given the limited spatial and temporal scope of the gridded emission and concentration fields, the direct use of these aspects of the dataset in future studies may be somewhat limited. Put differently, unless any follow-up studies requiring SBTEX data specifically focus only on this 5-month period in 2012 over this specific area, I expect to see little follow-up use of these portions of the dataset by other groups. The model evaluation R and python scripts used to generate the manuscript figures are nice and easy to follow for anyone familiar with these languages and could easily be adapted to perform similar analyses for other periods or regions.

Thank you for your comments. This study presents a method to identify possible missing STEX emissions in the national emission inventory and simulate the explicit chemical compounds that are not considered in the reduced chemical mechanism. There are very few studies that applied the chemical transport model with the reactive tracer method to simulate SBTEX concentrations for state-level range. Besides, the same simulation period (May to September 2012) as the Texas Commission on Environmental Quality (TCEQ) State Implementation Plans (SIP) ozone model (TCEQ, 2015) was selected to verify our CAMx core modeling results for the targeted species (O3, NO2, and formaldehyde) that related to oxidants (O3, NO3, OH radical).

Subsequently, this study carefully evaluated our modeling results with USEPA 87 surface ozone monitor sites and 46 AMTIC sites data for May to September 2012. The complete whole 2012 Adj case SBTEX concentration data (January to December) is developed, along with a monthly evaluation table. This 2012 SBTEX concentration data has been uploaded to Zenodo ( https://zenodo.org/record/8303346 ) with model evaluation tables (tables S9) and processing tools are also published with the data link.

This method's configuration and simulation processes are currently applied for additional modeling years from 2011 to 2016 for generating long-term SBTEX concentration data for supporting the National Institute of Environmental Health Sciences (NIEHS) cohort study. The Adj case SBTEX concentration results for 2011 and from 2013 to 2016 will be uploaded and follow the ESSD living data process policy once the QA/QC processes and model evaluation processes are completed.

With a strong background in emissions and air quality modeling over the U.S., the gridded data sets are usable and understandable, but without such a background, I would not consider them to be described and documented in sufficient detail. For example, the zenodo description lists these files as being in netCDF format, but omits the important fact that the spatio-temporal metadata information stored in these files follow I/O API conventions (https://www.cmascenter.org/ioapi/). These conventions, while often used in U.S. emissions and air quality modeling applications, are virtually unknown outside that community, making the interpretation of time and geolocation information of the values stored in these files impossible to many users. At a minimum, the documentation on both zenodo and within the article should make reference to the use of I/O API conventions and link to these conventions, but given the niche nature of I/O API and the learning curve associated with correctly interpreting its geospatial metadata, preferably the time and spatial coordinate and attribute conventions of these files should follow more widely used conventions such as netCDF-CF with explicit latitude, longitude, and time variables included in each file. Likewise, there is no explicit documentation on the meaning of variable names in the emission and concentration files. The "var_desc" attribute could be used to provide such documentation, but it currently just repeats the variable name without providing any further insight into, for example, the differences between "ALD2" and "ALD2_PRIMARY" which are not likely to be known to most readers / users. Documentation should be provided for all variables in all datasets. Because of these shortcomings in metadata and variable documentation, I do not consider the gridded datasets in their current form to be of high enough quality.

Thank you for the comments. We have converted the Adj case file to netCDF-CF format by using Python PseudoNetCDF (https://github.com/barronh/pseudonetcdf) and have uploaded the next version with detailed variable descriptions to Zenodo. For improved readability, we have also prepared a "comma-separated values (csv)" file containing hourly data for individual SBTEX

species in the Gulf Region for 2012. Additionally, we have included "Readme" files with those uploaded data on Zenodo (https://zenodo.org/record/8303346) (Readme_for 2012_SBTEX_conc_4km.txt, and Readme_for_2012_emission_12km.txt).

I am confused by the description of RTRAC as a "post-process feature" / "post-analysis feature", and "post-processing step" in the main article and supplement, and its depiction as a separate box in Figure S1. The RTRAC documentation in the CAMx user guide describes it as a built-in probing tool that, if enabled for a given CAMx simulation, is being applied simultaneously with the base model, and not after the CAMx simulation. Moreover, the description in the text and Figure S1 state that RTRAC tracer concentrations are affected by emissions and physical and chemical decay, but do not mention transport (advection and diffusion). Are the RTRAC SBTEX species not transported from their emission sources? Based on the CAMx user guide, I think they are, but based on the RTRAC method description provided in this article, I was left with the impression that they weren't.

Thank you for correcting this part. Yes, you are right, the RTRAC considers the physical transport processes, and it is probing tool. We replace the "post-process" with "probing tool", and correct the RTRAC description in manuscript (line 252-257) to embrace the transportation process. Now it reads:  The "Reactive Tracer" is a probing tool in the CAMx modeling system to simulate SBTEX concentrations. Along with the physical transport processes (diffusion and advection) and decay processes like wet and dry deposition same as the core model, there is the second-order chemical reduction rate $r$ that is calculated using the oxidants (Ozone, OH, $NO_3$) concentrations $[Ox]$, the SBTEX concentrations $[Tr]$, and the rate constants of reactions $k_{Tr+Ox}$ (Eq.3).

The method description in Section 2.2 of the main article should be improved as it would be critical for any user trying to replicate the CAMx-simulated concentration fields. The details of the air quality model description provided in Section 2 of the supplement should be included in Section 2.2 of the main article (e.g. information on the version of WRF, photolysis rates, and the generation of boundary conditions). There appears to be a contradiction between the main manuscript and the supplement about how WRF fields were prepared for CAMx – the main manuscript states "They were converted to SMOKE- and CAMx-ready gridded hourly meteorology through the Meteorology Chemistry Interface Processor (MCIP)" while the supplement states that "They are converted to the CAMx-ready format using the WRFCAMx version 4.8.1 program developed by the CAMx development team".

Thank you for your valuable comments. We apologize for any confusion, and we have made edits to section 2.2 to provide additional details about the air quality model in section 2.2.1. Regarding WRFCAMx and MCIP, we have revised line 220, and now it reads: "The WRF output data were transformed into SMOKE-ready gridded hourly meteorology through the Meteorology Chemistry Interface Processor (MCIP). Emissions sectors modulated by meteorology, such as onroad (Choi et al., 2014;Lindhjem et al., 2004) and biogenic, were estimated using the MCIP gridded hourly

meteorology. The USEPA's 2012 daily total wildfire emissions (ptfire) estimated by SMARTFIRE2 (USEPA, 2015) were also incorporated (USEPA, 2021). Additionally, the WRF meteorological data were converted to CAMx-ready meteorological data by using WRFCAMx (RAMBOLL, 2020) for the CAMx model input."

The method description should also define "flexi-nesting", a term not likely to be familiar to most readers or users of the dataset. Based on my read, both meteorology and emissions were processed for a 12 km grid, and the 4 km "flexi nest" grid was solely defined during the CAMx simulation with CAMx interpolating 12 km inputs to that finer grid during the simulation. Or were there any inputs actually prepared for the 4 km grid? (the distributed emissions dataset is for 12 km). If none of the inputs were prepared for the 4 km grid, what is the rationale for having CAMx interpolate 12 km inputs to that finer grid?

Thank you for this comment. For the area emission source sectors, there are no additional inputs needed for 4 km × 4 km flexi-nesting domain but the CAMx can interpolate coarse 12 km × 12 km gridded datasets internally. However, the points sources (ptegu, ptnonipm, pt_oilgas, and ptfire) are processed independently with their actual stack locations and their parameters (stack height, outlet temperature, outlet gas speed, and diameter…etc) to enhance their vertical allocations in any modeling domain. Therefore, the CTM model can enhance the horizontal and vertical spatial representations of ambient air pollutants compared to the 12km × 12 km domain. The example figures show a better spatial representation of SBTEX concentrations over the same modeling region between 12 km × 12 km (a), and 4 km × 4 km (b) for 2012.01.01 hour 17 GMT. The flexi-nesting method also allows users to efficiently provide accurate and consistent boundary conditions of ambient air pollutant concentrations for the 4km x 4km modeling simulations. We have edited the line 234-236 and clarified this information. Now it reads "The point source emissions are processed independently with their stack locations in the model domain and considering the plume-raising effect by stack parameters. As a result, the model spatial allocations can be enhanced through the flexi-nesting method."

[Figure]

I also cannot reconcile the fact that CAMx RTRAC simulations apparently were performed for that 4 km grid (implying that having 4 km concentration fields is desirable) with the fact that for model evaluation, the 4 km grid results are aggregated back up to 12 km by considering all nine grid cells surrounding a monitor (implying that modeled gradients at 4 km are not expected to be realistic).

Thank you for the comments. the evaluation method is based on USEPA's evaluation process (USEPA, 2006), the 4 km × 4 km consider the monitoring sites located within the grid cell with surrounding 8 grid cells; therefore, these 9 grid cells average are not the same as 12 km × 12 km domain.

The structure and language of the article and supplement are generally good, but there are a number of instances with somewhat awkward wording, with a few examples noted in the specific comments below. During revision, the article would benefit from a careful editorial review.

Thank you for your comments. We followed your specific comments listed and have edited below:

Specific Comments:

Line 49: Here and in all subsequent references to (USEPA, [year]) in the main article and supplement: please double check that all EPA citations listed in the references section actually have a publication year associated with them. At first glance, only the "Guidance on the Use of models and other analyses for demonstrating attainment of air quality goals for ozone, PM2.5 and Regional Haze" [which was actually updated in 2018] and "2014 Fire NEI Workshop Emissions Processing-SmartFire Details" references currently do, and yet there are many "(USEPA, [year])" citations throughout the text that don't have a clear connection to an item in the reference list.

Thank you for the comments. We corrected the issues in the reference list and citations.

Lines 108 – 110: see main comments above, is the reactive tracer function really a "post-process"?

Thank you, we have edited the reactive tracer in method section. it reads (line 108-110) "Besides, the reactive tracer function, which is one of the CAMx probing tools, allows the model to explicitly simulate SBTEX concentrations."

Lines 136 – 140: The first sentence says emissions "from certain facilities" are collected by EIS and then used to develop the NEI while the last sentence then says the NEI has "emissions … from all types of emission sources". Please clarify.

This paragraph explains that USEPA NEI data can have both Hazardous Air Pollutants (HAPs) and Criteria Air Pollutants (CAPs) emission data.  However, the HAPs data is usually voluntary reported by state agencies. We removed the unclear description that could confuse the readers, and now it reads (line 136-142):"The NEI is a national database providing comprehensive annual air emission estimates for both criteria air pollutants (CAPs) (e.g., CO, $NO_X$, $SO_2$, $NH_3$, VOC, and $PM_{2.5}$), and HAPs (e.g., benzene, acetaldehyde, formaldehyde, xylenes, styrene, and more) from all types of emissions sources (e.g., point, nonpoint, and mobile). While CAPs emissions are reported by the state agencies is mandatory, the report of HAPs is usually voluntary. Consequently, only the limited set of HAPs have been reported to the USEPA, and their spatial coverage can vary significantly by source type (e.g., industrial, vehicles) and region (e.g., county and state) (Strum et al., 2017). "

Line 139: CO is misspelled

Thank you, we have corrected the typo.

Line 145 – 146: Does this distinction indeed apply to the NEI, or more broadly to an emissions modeling platform that includes the NEI but also tools like SMOKE and speciation databases? Specifically, does the NEI actually include "model surrogate" species, or are they only computed from NEI information (VOC and/or HAP explicit) during emissions processing for a specific chemical mechanism in a specific modeling platform?

Thank you for correcting this part, the NEI can have CAPs and HAPs, but doesn't have model surrogate species, which is generated by VOC speciation profiles during the chemical speciation processing step in the SMOKE modeling. We edited this description, and now it read (line 143-148): The VOC emission species generated by SMOKE from NEI have three types, "model surrogate", "model explicit", and "HAPs explicit" species. The "model surrogate species", such as XYL (Xylene and other poly-alkyl aromatics), TOL (Toluene and other mono-alkyl aromatics), and PAR (paraffin carbon bond), are calculated by VOC speciation profiles in emission model

platform and used to predict ozone and secondary organic aerosol (SOA) in the CTM but not for individual HAPs emission and simulation.

Line 193: should "Evoc" be "Evoc,s,f"?

Thank you, it has been corrected. Now it reads (line 195) :"$E_{voc,s,f}$ is the CAP VOC emission for the SCC in the county."

Lines 198 – 199: unclear writing. Perhaps "Only the emission sources for which the sum of all HAPs is zero (sum Eisf = 0) are considered as the "without HAPs" group" instead?

Thank you, we have edited this (Line 200). Now it read "Only the emission sources for which the sum of all HAPs is zero ($\sum_i E_{i,s,f} = 0$) are considered as the "without HAPs" group."

Line 218: "The emissions sectors modulated by meteorology" instead of "meteorology-induced"?

Thank you, we have edited this sentence. Now it read (line 221) "The emissions sectors modulated by meteorology, such as onroad (Choi et al., 2014;Lindhjem et al., 2004) and biogenic, were estimated with the MCIP gridded hourly meteorology."

Line 221: "used" rather than "imported"?

Thank you for your edit. We have edited this sentence and now it read (line 223-224): "The USEPA's 2012 daily total wildfire emissions (ptfire) estimated by SMARTFIRE2 (USEPA, 2015) were also incorporated (USEPA, 2021)."

Lines 224 – 230: See main comment above regarding clarification of RTRAC.

Thank you, we incorporated your comments and have edited all RTRAC description part.

Line 245: please define "long-term" and justify why the focus was on long-term evaluation.

To accurately reflect the time period, we replaced "long-term" with "5 months (May to September 2012)" in line 277.

Lines 252 – 254: this suggests that only high outliers (2*IQR above Q3) were removed while no screening was performed for low outliers. If so, please state explicitly and justify.

The USEPA has already conducted QA/QC for the AMTIC data. As a result, all values from AMTIC are considered genuine data. The presence of high outliers is attributed to fugitive emission events or random releases of VOC from the oil and gas industries (Couzo et al., 2012). These events are challenging to represent accurately using regulatory emission models and air quality models. However, these exceptionally high data points have a notable impact on the model's performance, including correlation coefficients and mean bias. Consequently, only the high outliers (random VOC emission events) have been eliminated. We have addressed this in the manuscript, which now reads (lines 283-288): " The USEPA conducted QA/QC for the AMTIC data, which contain values that are exceptionally high due to unpredictable industrial VOC release events (Couzo et al., 2012). These events are beyond the regulatory emission counting; thus, the model cannot capture those unpredictable events, particularly in petrochemical, oil, and gas industrial areas. Therefore, this study removed outliers (those beyond twice the interquartile range (2×IQR) above Q3) for better evaluating the model performance in simulating the SBTEX concentration in general."

Line 265: I believe this should be "vessel", not "vehicle"

Thank you for correcting this in the manuscript. Now it reads (line 300): "commercial marine vessel (cmv)"

Line 290: change "some rural area" to "some rural areas"

Thank you for correcting this in the manuscript. Now it reads (line 325): "some rural areas in Texas"

Line 301: suggest "reduce" instead of "mitigate"

Thank you, we have edited this in the manuscript. Now it reads (line 336): The inclusion of the missing sources will slightly reduce the emission variation across a day,

Lines 318 – 324: There are some awkward sentences in this paragraph, please review and revise.

We have edited the manuscript (line 402-411). The revised version is as follows: "The spatial distribution patterns of individual SBTEX compounds exhibit similarities due to shared emission sources, except for styrene. Styrene primarily originates from ptnonipm, while other species predominantly arise from vehicle emissions and wildfires. benzene (max: 1.06 ppb), toluene (max:

1.01 ppb), and ethylbenzene (max: 0.16 ppb) reach their highest concentrations in Houston, reflecting their significant emissions. Further, xylenes (0.78 ppb) originate from sources in Shreveport. Remarkably elevated concentrations of styrene (reaching 1.97 ppb) are conspicuously identified proximal to Lake Charles, a locale characterized by an abundant emission of styrene from non-Electricity Generating Unit point sources, which have been absent in the original NEI records."

Lines 342 – 345: The first line states that it's mostly meteorology, but then the third line says both meteorology and more active daytime chemistry and the associated chemical loss of these tracers act to decrease daytime tracer concentrations. Review and revise.

We edited the manuscript (line 428-432). Now it reads: "In general, the diurnal variations of SBTEX concentrations are primarily influenced by various factors (such as ventilation, emissions, diffusion, deposition, and chemical reactions). These variations typically manifest with lower concentrations during the daytime compared to nighttime due to increased ventilation, diffusion, and chemical loss, even though emissions are about 4 times higher during the daytime, as presented earlier (Fig. 4)."

Line 346: "sensitivity of the concentration to emissions" instead?

We updated the manuscript (line 432). Now it reads: "Diurnal meteorological and emission patterns suggest more sensitivity of the concentrations to the emissions during nighttime than daytime, implying that implementing emission controls to reduce the concentrations at night would be most effective."

Line 347: Are nighttime exposures when people are mostly at home a concern, and should lowering concentrations during that time period be a priority?

The higher concentration period should be given priority. However, this statement also depends on the locations (residential, industrial, or urban areas) and human activities. We believe that individuals with more nighttime activities, such as night workers in industrial areas or airport employees working at night, may be more susceptible to the effects of these toxicants in urban environments. Furthermore, houses lacking proper insulation systems or inhabited by low-income populations do not use air conditioning, which could directly expose to these high-concentration toxicants. This study offers a comprehensive SBTEX concentration map for the Gulf region to identify hotspots, and its findings can be applied in future epidemiological studies on human exposure. To address the review concern. We addressed in manuscript (line 484-486), now it reads"

The high SBTEX concentration during nighttime affects individuals who engage in more nighttime activities or reside in houses lacking isolation of outdoor air."

Line 356: Suggest not starting this sentence with "in contrast" because it actually says that the level of peak SBTEX is the same for both locations. Instead, start the next sentence "In contrast to Channelview, the peak concentration at Bayland Park occurs …."

Thank you for this suggestion; we edited this part (line 443-448). Now it reads "The second case, Bayland Park (Latitude: 29.69, Longitude: -95.49), located nearby at the western side of Houston, presents the same level of peak SBTEX concentration (about 12 ppb) (Fig.9a) as Channelview city. In contrast to Channelview, the peak concentration of Bayland Park occurs at traffic rush hour (LT 7:00 to 8:00), contributed mostly by Toluene (53%) and Xylene (23%) (indicating the mobile vehicle sources) rather than Benzene (18%)."

Line 361 – 362: awkward wording of the first part of the sentence, review and revise.

Thank you, we edited this in the manuscript (line 448-450). Now it reads "Meanwhile, the adjusted industry emission sources, as presented in table S5, play a significant role in driving the peak concentration (0.4 ppb) in Channelview city (Fig. 8b), yet exhibit a reduced impact on Bayland Park (Fig.9b), where is far from the industry area."

Line 371 – 372: change "composition of" to "contribution from"?

Thank you, we edited this in the manuscript. Now it reads (line 458): "The missing emission sources (Fig. 10b) will further enhance the peak concentration by 2 ppb at LT 5:00 - 8:00, with the largest chemical contribution from Toluene (about 70 - 85%), followed by Styrene (about 7 - 20%) associated with the industrial sources."

Line 420: "applied" rather than "implemented" since no modifications to off-the-shelf CAMx code were made?

Thank you, we edited this in the manuscript. Now it reads (line 466) : "Then the state-of-the-science chemical transport modeling system, CAMx, was applied to generate the high……"

Line 429: "underestimated" instead of "missed"?

Thank you, we edited this in the manuscript. Now it reads (line 474): "……but were substantially underestimated in the original NEI data……"

Line 493 – 495: awkward wording, review and revise.

Thank you, we edited the manuscript (line 545-548). Now it read: We want to thank the National Institute of Environmental Health Sciences (NIEHS) support this study, the research project is "Neurological Effects of Environmental Styrene and BTEX Exposure in a Gulf of Mexico Cohort" (Grant No. R01ES031127). We also appreciate the grant support including……

Line 504: Ramboll, not Ramball.

Thank you for correcting this typo in manuscript (line 557).

RC2

This study uses observations and modeling to create an emission map of five hazardous air pollutants (HAPs) for May-Sep 2012 in the U.S. Gulf region. This map includes emissions that are missing in the US EPA emission inventory. High emissions were reported in the literature for these five HAPs in the Gulf region and they pose health risks, therefore an accurate emission inventory is beneficial. The analysis methodology is sound. However, I have some concerns listed below.

Major comments:

The authors should put more effort into justifying the use of various datasets. The current time coverage of the emission map generated is only five months in 2012. Why do authors not use more up-to-date datasets to generate an emission map for longer and more recent time periods? The AMTIC ambient measurements are available until 2023, and US EPA NEI has the most recent version for SBTEX emissions in 2023. So I'm confused why the authors used 2012 ambient measurements and 2011 NEI. If authors find 2012 emissions very interesting, I encourage them to extend the method to the most recent (at least until 2022) to have a 10-year monthly emission map for the Gulf region. This will make the study more interesting to the community.

Thank you for your comments. There are very few studies that have applied reactive tracers to simulate SBTEX concentrations. Therefore, this study meticulously verified the model processes. We selected the same simulation period (May to September 2012) as the Texas Commission on Environmental Quality (TCEQ) State Implementation Plans (SIP) ozone model (TCEQ, 2015) to validate our CAMx modeling results for oxidants species (OH, $O_3$, and $NO_3$). We accomplished this by comparing concentrations of ozone, NO2, and formaldehyde. Those species are connected to oxidants (O3, NO3, and OH radicals) in the atmosphere and contribute to the chemical decay of SBTEX. Additionally, the model results were evaluated for ozone performance using USEPA surface air quality stations. Furthermore, the SBTEX concentrations derived from the reactive tracer process were compared with observational data from AMTIC.

Moreover, the emission data used to generate the concentration map were sourced from NEI, with the latest available data year being 2019 (released in Sep. 2022, https://www.epa.gov/air-emissions-modeling/2017-2019-air-emissions-modeling-platforms). Therefore, we are unable to update the SBTEX concentration map to 2022 now. However, this study is in collaboration with the National Institute of Environmental Health Sciences (NIEHS) SBTEX exposure cohort study for the Gulf region for 2011-2016. We have uploaded the 2012 Adj case concentration data ( https://zenodo.org/record/8303346 ), confirming its monthly performance (correlation coefficient and normalized mean bias for model performance, table S9). The Adj case SBTEX concentration results for the other years (2011, 2013-2016) will be uploaded and will follow the ESSD living data process policy once their QA/QC processes and model evaluation are completed.

Specific comments:

Line 96-97: why dispersion models cannot support regional scale application? Also add references.

The dispersion models are employed to simulate air pollutant concentration at the downwind location (USEPA, 2023). The dispersion models are driven by an hourly single wind field (please see "https://gaftp.epa.gov/Air/aqmg/SCRAM/models/preferred/aermod/aermod_userguide.pdf" page 3-134 surface meteorological data input). The dispersion model considers only a single wind field and very simple chemistry process for all receptors (USEPA, 2022). In contrast to dispersion model, chemical transport models (CTM) are the Eulerian models, which consider multiple grid cells (3 dimensions and time), and each grid cell has its own meteorological input. Consequently, the CTM model can simulate air quality on state-level scale with the chemistry processes. We added the references for this description and have edited. Now it reads (line 96-98): "These dispersion models, however, only account for exposures at a smaller spatial scale (USEPA, 2022, 2023) thus cannot support regional scale (e.g., state-level) application."

Line 129: provide reference for NEI 2011.

Thank you, we added the reference to the manuscript.

Line 130-133: explain the difference between NEI vs. TRI vs. EIS.

NEI (National Emission Inventory) is the emission database that considers all anthropogenic emissions including criteria air pollutants (CAPs) (e.g., CO, $NO_X$, $SO_2$, $NH_3$, VOC, and $PM_{2.5}$) and hazardous air pollutants (HAPs) (e.g., Benzene, Acetaldehyde, Formaldehyde, Xylenes, Styrene, and more). TRI (Toxics Release Inventory) is the toxics (or HAPs) emission inventory but only for special emission sources especially the larger HAPs emission sources. EIS (Emission inventory System) is the interface for state agencies to report the emission data including CAPs and HAPs to USEPA. We edited this paragraph for clarification. Now it reads (line 128-133) "In this study, the 2011 version 6 NEI was applied as the base emission inventory (USEPA, 2021). Subsequently, the SMOKE modeling system was employed to produce hourly gridded emissions of SBTEX across the Gulf modeling region for the year 2012. These SBTEX emissions were utilized in conjunction with the CTM model and a reactive tracer function to generate the SBTEX concentration map. In the end, the USEPA Ambient Monitoring Technology Information Center (AMTIC) data were employed to evaluate the model performance in simulating SBTEX concentrations."

Line 140: lowercase for "Styrene"

Thank you for correcting this in the manuscript. For consistency in the manuscript, we have changed all chemical compounds full name to lowercase.

Line 197-198: this means that their emissions are assumed to be correct? Having no missing emission doesn't mean it is correct. Uncertainty and evaluation against observations should also be conducted for them.

This study assumed that when VOC and their associated HAPs emissions are present in the NEI, we consider them to be consistently estimated. It's accurate to acknowledge that this doesn't necessarily imply their correctness, but our assumption is that HAPs have been reported alongside the VOC emissions. For instance, using SCC code "2310021010," which corresponds to "Industrial Processes; Oil and Gas Exploration and Production; On-Shore Gas Production; Storage Tanks: Condensate," we assumed that the same emission processes will result in similar VOC and HAPs compositions. Hence, when NEI HAPs data share the same SCC code "2310021010" but display all HAPs as zero, we interpret this as missing HAPs emissions. We are aware of the uncertainties inherent in this approach; therefore, in section 3.2 Model performance, we compared the CTM model's results with observational data to assess these uncertainties. Furthermore, we have discussed these uncertainties in our conclusion and discussion.

Section 2.2: what is the spatial and temporal resolution of CAMx RTRAC?

The spatial resolution is 4 km × 4 km, and the temporal resolution is hourly. We edited sector 2.2 Model configuration part to show more model setup details in the manuscript.

Line 218-220: should include more details about meteorology-induced emissions sectors, biogenic and wildfire emissions in the main manuscript, I see they are now in SI.

We have edited sector 2.2 Model configuration part and add more basic model details to this sector.

Figure S1: I recommend including it in the main manuscript.

Thank you for this comment. We have added this figure as figure 2 in sector 2.2 in manuscript.

Section 3: should provide more specifics on the confidence intervals, and uncertainty range of the emission rates or concentration reported.

We acknowledge the significance of data uncertainty. Regarding the uncertainty related to emission data, it's unfortunate that the USEPA has only provided an annual emission database and

emission model platforms. There are no report available detailing emission uncertainty ranges that have been estimated by the USEPA and published in NEI or incorporated into the emission model platforms. Certain studies have employed their own methods to estimate potential emission uncertainties (Frey, 2007; Yan et al., 2014; Zhao et al., 2017). However, this aspect falls outside the scope of our current research. Regarding the concentration part, which is the focus of our study, section 3.2 of the results presents the model evaluation for SBTEX using correlation coefficients and normalized mean bias to illustrate the model's performance. These evaluation methods have been developed by the USEPA (USEPA, 2006) and the air quality model community.

Figure 2, 3: should include time periods in the figure legend.

We edited the Figure caption (now Figure 3) with more details for the time periods. Now it reads "Spatial distribution of 2012 annual total SBTEX emission rates (ton/yr) of the modified emission inventory used in this work (a), and the location and amount of emissions that were added to the NEI (b)."

Section 3.3: as model validation should come as section 3.1.

Thank you for the comment. We put model result evaluation into section 3.2 (previous version is 3.3), and SBTEX concentration pattern description into 3.3 (the previous version is 3.2).

Table 2: (1) should provide such information for each site, each month and individual pollutant to show how they differ. (2) it seems that including missing emissions ("Adj") does not improve R values, any reason for that?

(1) Due to the fact that many sites only provide weekly data (with one day average per week), we have incorporated a monthly model evaluation table for correlation coefficient and normalized mean bias across all sites in tables S9a and b.

(2) The model outcomes are the result of multiple physical and chemical processes, encompassing meteorology, chemistry, emissions, and deposition. The principal factors contributing to daily variations are driven by emissions, chemical decay, and meteorological conditions. In this study, we only adjusted the annual total STEX emission. The annual STEX emission (measured in ton $yr^{-1}$) within NEI is processed by SMOKE to generate gridded hourly data utilizing monthly, weekly, daily, and hourly temporal profiles. These temporal profile divisions are the same as the Base case (for example, the weekday rush hour or reduced weekend traffic emissions in urban areas). The R values are primarily influenced by concentrations ranging from low to high. Consequently, there is not a significant enhancement in R-values within the Adj case. We have edited the conclusion and discussion part for addressing this in manuscript. Now it read (line 492-498): "This study is mainly based on the bottom-up NEI dataset, thus the uncertainties in the original NEI emissions and SMOKE process influenced on this study. For example, despite our implementation of

imputation for the HAPs annual data, the emission activity within hourly, daily, and monthly temporal profiles as well as parameters (e.g., emission rates, and compositions) also remains unchanged. The emergency emissions from unreported flaring (such as final treatment equipment) or leakage events that have not been considered in the original NEI, also not included in this study."

Figure S8: there is a > 2-factor difference between observation and model for BTES from 10 pm to 8 am, this should be discussed.

Thank you. We added a paragraph to discuss the details in the manuscript. Now it reads (line 387-397): "Because only few sites have hourly data, this study compared the diurnal variation of SBTEX concentrations for Houston industry area (using data from only three monitoring sites) in Fig. S8. The hourly data shows that benzene, ethylbenzene, and styrene are overestimated (NMB for benzene is 69%, ethylbenzene is 36%, and styrene is 27%) during nighttime hours (LT 21:00 to 6:00). Toluene is underestimated at nighttime (NMB is -45%), whereas xylenes closely aligns with the observed data (-16%) range. On the other hand, all species experience underestimation during daytime hours (NMB for benzene is -25%, Toluene is -65%, Xylenes is -51%, ethylbenzene is -46%, and Styrene is -82%) (from LT 10:00 to 17:00). Such results indicate that the hourly emission rate may overestimate during nighttime but underestimate during the daytime in Houston industry area." The conclusion and discussion part also explained the uncertainties of our method (line 489 to 502).

Figure 8: color of diamonds are observational data? It is hard to compare observation with model results in this figure.

Thank you for providing these comments. We have revised this figure (now labeled as Figure 5) by using larger diamond shapes and zooming in on the details. The diamond colors correspond to the observational results, and they are intended to closely match their background color (representing model results). This alignment signifies that the model results closely match the observational data and capture concentration gradients. Additionally, Figure 6 illustrates a one-to-one data comparison of the data presented in Figure 5 for all SBTEX species.

Section 4: should compare the new emission map to NEI and discuss how US EPA can improve their estimates, potentially also provide a correction factor for NEI to match the new emission map.

Figure 3b illustrates the difference for STEX emission map, and table S5 display the emission difference by states and emission sectors. Furthermore, both table S6 and table S7 provide the detailed breakdown of the potential missing emissions by SCC. The result shows that the non-point oil and gas sector (np_oilgas) have largest difference, and point source emission other than electricity generation unit (ptnonipm) have the second largest possible missing emission. Detailed

comparisons between the NEI and new HAPs emission data are available in Supplementary Document Section 3, titled "Missing HAPs Emission from oilgas and ptnonipm. In the correction factor aspect, we did not generate a correction factor specifically for the NEI. Instead, we employed the HAPI program to determine the ratio of HAPs to VOC emission data by NEI, enabling us to generate SCC-level HAPs to VOC ratios. Subsequently, we utilized this ratio to perform the imputation process.

References: some references missed the publication year or the year of last access date.

Thank you for correcting this, we have edited the revision and fixed this issue.

In Supplementary material: line 67: hydroperoxyl radical is HO2, OH is hydroxyl radical.

Thank you for correcting this, we fixed this issue.

**Reference list:**
Choi, K.-C., Lee, J.-J., Bae, C. H., Kim, C.-H., Kim, S., Chang, L.-S., Ban, S.-J., Lee, S.-J., Kim, J., and Woo, J.-H.: Assessment of transboundary ozone contribution toward South Korea using multiple source–receptor modeling techniques, Atmospheric Environment, 92, 118-129, https://doi.org/10.1016/j.atmosenv.2014.03.055, 2014.

Couzo, E., Olatosi, A., Jeffries, H. E., and Vizuete, W.: Assessment of a regulatory model's performance relative to large spatial heterogeneity in observed ozone in Houston, Texas, Journal of the Air & Waste Management Association, 62, 696-706, 10.1080/10962247.2012.667050, 2012.

Lindhjem, C., Chan, L., Pollack, A., Corporation, E. I., Way, R., and Kite, C.: Applying Humidity and Temperature Corrections to On and Off-Road Mobile Source Emissions, 13th International Emission Inventory Conference, Clearwater, FL, 2004.

CAMx7.00 User's Guide https://camx-wp.azurewebsites.net/Files/CAMxUsersGuide_v7.00.pdf, 2020.

2011 Version 6 Air Emissions Modeling Platforms: https://www.epa.gov/air-emissions-modeling/2011-version-6-air-emissions-modeling-platforms, access: April, 04, 2022, 2021.

User's Guide for the AMS/EPA Regulatory Model (AERMOD): https://gaftp.epa.gov/Air/aqmg/SCRAM/models/preferred/aermod/aermod_userguide.pdf, access: August 15, 2023, 2022.

Air Quality Dispersion Modeling: https://www.epa.gov/scram/air-quality-dispersion-modeling#:~:text=Dispersion%20modeling%20uses%20mathematical%20formulations,at%20selected%20downwind%20receptor%20locations., access: August 15, 2023 2023.